neuroscience/behaviour

handedness, laterality, hand skills, behaviour, sex effect, ALSPAC

**Author for correspondence:**
Silvia Paracchini
e-mail: sp58@st-andrews.ac.uk

# Different laterality indexes are poorly correlated with one another but consistently show the tendency of males and females to be more left- and right-lateralized, respectively

Carlos Buenaventura Castillo[1,2], Andy G. Lynch[1,2] and Silvia Paracchini[1]

[1]School of Medicine, and [2]School of Mathematics and Statistics, University of St Andrews, Saint Andrews, Fife, UK

CBC, 0000-0001-8328-4640; AGL, 0000-0002-7876-7338; SP, 0000-0001-9934-8602

The most common way to assess handedness is based on the preferred hand for writing, leading to a binary (left or right) trait. Handedness can also be assessed as a continuous trait with laterality indexes, but these are not time- and cost-effective, and are not routinely collected. Rarely, different handedness measures are collected for the same individuals. Here, we assessed the relationship of preferred hand for writing with four laterality indexes, reported in previous literature, derived from measures of dexterity (pegboard task, marking squares and sorting matches) and strength (grip strength), available in a range of $N = 6664$–8069 children from the ALSPAC cohort. Although all indexes identified a higher proportion of individuals performing better with their right hand, they showed low correlation with each other (0.08–0.3). Left handers were less consistent compared to right handers in performing better with their dominant hand, but that varied across indexes, i.e. 13% of left handers performed better with their right hand on marking squares compared to 48% for sorting matches and grip strength. Analysis of sex effects on the laterality indexes showed that males and females tend to be, on all measures, more left- and right-lateralized, respectively. Males were also over-represented among the individuals performing equally with both hands suggesting they had a higher tendency

to be weakly lateralized. This study shows that different handedness measures tap into different dimensions of laterality and cannot be used interchangeably. The trends observed across indexes for males and females suggest that sex effects should be taken into account in handedness and laterality studies.

# 1. Introduction

Worldwide, the vast majority of people (roughly 90%) prefer using the right hand for most tasks in contrast to a minority of about 10% who prefer the left hand [1,2]. The 1 : 9 left to right ratio for hand preference is a feature specific to humans and is observed across populations. Language acquisition, a function characteristic of humans, is also a strongly lateralized trait with left hemisphere dominance observed in most individuals [3]. A link between handedness and language, mediated by hemispheric specialization, was first made with the Wada test [4]. This link has since provided the foundations for assuming a role of brain asymmetries in human evolution [5]. In this context, handedness which is the most obvious and accessible lateralized trait, has been investigated for association with cognitive skills, personality and psychiatric disorders [6–8]. However, inconsistent findings have revealed complex patterns and the cause/effect relationship between handedness, brain asymmetries and disorders remains unexplained and debated [9].

Intuitively, handedness is a binary category and most studies rely on the assignment of participants to a left/right-hand status based on preferred hand for writing. A binary classification has many advantages, such as convenience and cost-effectiveness, but it has been argued that it is not sufficient to capture a general handedness dimension [10,11]. The Edinburgh Handedness Inventory (EHI) provides a solution by assessing hand preference scores on a series of tasks [12], including items such as brushing teeth, which are not expected to be influenced by cultural pressures. In fact, it is well established that environmental pressures might force the use of the right hand for writing in left handers [13]. This phenomenon was mainly observed in past generations, and still applies to some cultures [1,14]. The EHI, similarly to other handedness questionnaires, leads to scores with a J-shaped distribution, indicating that the majority of people present an overall right- or left-hand preference with a few individuals in between.

Instead, handedness indexes are derived by comparing performance of the two hands in carrying out skilled tasks. Most typically, these tasks measure speed and dexterity, such as the Annett pegboard [15], marking squares and sorting matches tasks [16]. The grip-strength test, instead, measures differences in manual strength [17]. Test-retest correlations show that these tasks lead to reliable scores [18,19]. Handedness indexes are derived by comparing the scores of the right (R) and left (L) hand (see Material and methods) and lead to continuous distributions with positive means indicating a higher number of individuals who perform better with the right hand. This observation suggested that hand preference and laterality indexes are roughly measuring the same thing. For example, hand preference was shown to correlate with indexes derived from peg moving [20] and finger-tapping [21] tasks. Bishop [22] also proposed a model consistent with the idea that the relative proficiency between the two hands directly determines hand preference. Porac & Coren [23] questioned this assumption after reviewing studies that tested individuals both on performance and preference tasks. They found an average agreement of 74% but with variation across tests. In spite of this variability, differences across laterality indexes have not been fully addressed. The grip strength task was one of the first measures used to assess asymmetrical skills between the two hands [16,24]. The grip task has since been used in different handedness studies [25–27] and also inspired Annett's right shift model based on the idea that handedness is a continuous and normally distributed variable [15]. The peg moving task used by Annett compares hands on a dexterity task and has been extensively used in the handedness literature [18,26,28]. In spite of the different domains (i.e. strength versus dexterity), different performance tasks show that right handers are stronger with their preferred hand, whereas left handers tend to be more inconsistent [29–31]. Very few studies collected multiple measures on the same participants and even fewer of them assessed the relationships across indexes. Annett showed that the laterality indexes derived from the peg moving task correlated with other laterality indexes (i.e. marking squares, dotting between targets, line drawing and hole punching) with an average of 0.5 and a range of 0.38–0.65. However, correlations were not presented across all indexes. Brown et al. [26] assessed what task, out of two pegboard versions (i.e. moving pegs and placing pegs), finger tapping and grip strength, best predicted hand preference. They also reported correlation across indexes which, with the exception of the two pegboard tasks ($r = 0.633$), were modest

and ranged between 0.01 and 0.3. A more recent study [32] showed a higher correlation (0.73) between the laterality indexes derived from the pegboard moving task and the Tapley and Bryden's circle marking task [18]. However, the correlation dropped to 0.14 when calculated within groups (dominant versus non-dominant hand). This is also consistent with the original Tapley and Brayden study which found that the correlation between preference and performance was 0.75 in the overall sample, but dropped to 0.17 in the right handers and 0.20 in the left handers. These data, typically collected in relatively small samples, highlight the high variability across studies, and indicate lack of a systematic assessment of performance measures. A difficulty to address this gap is the availability of adequate datasets.

While hand preference data are easy to collect in extremely large samples through self-reported questionnaire [33], performance tests are expensive and time-consuming, as they require one-to-one assessment by trained personnel. A solution is offered by birth and epidemiological cohorts which conduct very detailed assessments. For example, two laterality indexes (marking square and sorting matches) were analysed in over 12 000 children [16] from the National Child Development Study (NCDS) [34,35] . However, this study, which presented the scores as two separate plots for males and females, did not examine the correlation across the two tasks. Sex effects are also poorly investigated factors in the context of laterality indexes derived from performance measure. This is in contrast to the evidence that a higher prevalence of left-handedness has been consistently reported in males [36]. Sex effects have been studied for performance tasks but not on their derived laterality indexes. For example, males tend to have higher scores than females in strength tests [37] and females have been reported to perform faster than males with both their preferred and not-preferred hand on the peg task and other dexterity tests [38–41]. Some of the effects on dexterity have been suggested to be the results of smaller finger sizes in females compared to males, in tests like the pegboard [38,42,43] but this interpretation is not consistently supported [44]. Given the sex effects on hand preference and on performance task, a systematic analysis on the derived laterality indexes seems justified.

By taking advantage of the Avon Longitudinal Study of Parents and Children (ALSPAC), we aimed to address some of the current gaps in the handedness literature. We analysed different handedness indexes derived from peg placing, marking squares, sorting matches and grip strength tasks in a range of 6664–8069 children. All indexes showed a higher proportion of children performing better with their right hand but were poorly correlated with one another. Consistent with previous literature, we found a higher frequency of left-handedness in males. Sex effects on the laterality indexes showed that male and females tend to be more left- and right-lateralized, respectively. These data show that different handedness measures are not interchangeable but are similarly influenced by sex.

# 2. Material and methods

## 2.1. The ALSPAC cohort

ALSPAC is a longitudinal cohort representing the general population living in the Bristol area. The ALSPAC cohort consists of over 15 000 children from the southwest of England that had expected dates of delivery between 1st April 1991 and 31st December 1992 [45,46]. From age 7, all children were invited annually for assessments on a wide range of physical, behavioural and neuropsychological traits, including cognitive (reading and mathematics related) measures. Attendance to the annual assessment determined the availability of data for the measures used in this study.

## 2.2. Phenotypes

Self-reported hand preference for writing was collected at age seven. Four other tests previously used in the handedness literature were available in the ALSPAC dataset.

The pegboard task was conducted as part of the movement assessment battery for children to assess dexterity (Movement ABC; [47]). The child had to insert twelve pegs, one at a time, into a peg board, holding the board with one hand and inserting the pegs with the other, as quickly as possible. The task was carried out with the preferred and the non-preferred hand, after it had been described and demonstrated by the tester, and after a practice with each hand. The score corresponded to the time taken to complete the task with each hand.

The marking square and sorting matches tasks are a repetition of those used in the NCDS [48] and were collected at age 10 to assess laterality. Both tasks were first demonstrated by the tester and then the child had a practice. Scores were then collected from two measurements for each hand.

**Table 1.** Distribution of hand preference by sex.

| | right | left | total |
|---|---|---|---|
| male | 3530 (86.4%) | 558 (13.6%) | 4088 |
| female | 3562 (89.5%) | 419 (10.5%) | 3981 |
| total | 7092 (87.9%) | 977 (12.1%) | 8069 |

In the marking square task, the child is asked to make a short dash with a pencil on a piece of paper which has a grid made of rows of 20 squares. They are asked to start at the top left-hand side of the squared paper and move across it. When the first line is completed the child should move on the left side of the next row. The score corresponded to the number of squares that could be marked in 60 s and it was derived from the mean score of the two trials.

In the sorting matches task, the child is asked to move one match at a time across two boxes, one full and one empty using one hand only. The score corresponded to the time taken to transfer all the matches from one box to the other.

The grip test was assessed with a Jamar hand dynamometer at age 11 to measure strength. After one demonstration from the tester, the child was given the opportunity to practice. The child was encouraged to squeeze the apparatus as long and as strongly as possible. The measurements were taken for alternating hands for three times and starting with the right hand. The higher the reading (measured in kilograms), the stronger the grip. The mean from the three measurements for each hand was used.

Please note that the ALSPAC website contains details of all the data that is available through a fully searchable data dictionary and variable search tool http://www.bristol.ac.uk/alspac/researchers/our-data/. See electronic supplementary material for extracts of the documentation relevant to these tasks.

When scores for performance tasks were recorded for the 'dominant/non-dominant' hand, we used information about the preferred hand for writing at age 7 to define the scores for the left and right hand. When multiple trials were available a mean score across the trials was used. For all four performance tasks, we derived laterality indexes, namely PegQ, MarkQ, SortQ and GripQ, based on previous literature so that positive and negative scores corresponded to a better performance with the right and left hand, respectively [20,49].

## 2.3. Data analysis

All analyses and data visualization were performed using a number of packages (see full list in electronic supplementary material, table S1) R Studio v. 3.6.2 [50].

# 3. Results

## 3.1. Handedness measures

Analysis was conducted using data from the ALSPAC cohort. We assessed the frequency of hand preference for writing recorded as a self-reported measure when children were 7 years old ($N = 8069$, table 1). In total, 977 (12.1%) were left-handed. Consistent with previous literature [36], there was a higher proportion of left-handed males (13.6%) than females (10.5%). This measure did not identify any ambidextrous individuals.

Four laterality indexes (PegQ, SortQ, MarkQ and GripQ) were derived from manual tasks that measure the performance with both the right (R) and left (L) hand. For all indexes, a positive value indicates a better performance with the right hand (table 2).

MarkQ presents a bimodal distribution, consistent with previous reports [16], while the others are unimodal and well-approximated by a normal distribution (table 2; electronic supplementary material, figure S1). All indexes are leptokurtic and have a positive mean, indicating that the majority of individuals performed better with the right hand. PegQ and SortQ exhibit discrete behaviour near zero as expected from their construction (see electronic supplementary material, figure S1). MarkQ presented 13% negative scores, similar to the frequency of individuals who preferred writing with the left hand. The other indexes had a larger proportion of negative scores (24%–32%; table 2).

Most data were collected within a period of roughly three months around the target age for each test but the actual age range spanned across two years (table 2; electronic supplementary material, figure S2).

**Table 2.** Laterality indexes. Age was reported in days (as in electronic supplementary material, table S1), but presented here as months for easier interpretation. See supplementary material, figure S1 for the age distribution across the indexes and supplementary material, figure S2 for age effects on the indexes.

| index | formula | N | age (months) | | N trials/hand | index distribution | | | | % negative scores |
| | | | mean | s.d. | | mean | s.d. | skew | excess kurtosis | |
| --- | --- | --- | --- | --- | --- | --- | --- | --- | --- | --- |
| PegQ | 100*(L−R)/(L + R) | 6884 | 92.3 | 3.9 | 1 | 5.37 | 9.81 | −0.19 | 0.35 | 24.16 |
| MarkQ | 100*(R−L)/(L + R) | 7389 | 130.1 | 3.2 | 2 | 13.95 | 13.59 | −0.65 | 0.96 | 13.51 |
| SortQ | 100*(L−R)/(L + R) | 7366 | 130.1 | 3.2 | 2 | 2.58 | 7.13 | −0.07 | 0.18 | 32.65 |
| GripQ | 100*(R−L)/(L + R) | 6664 | 143.4 | 2.9 | 3 | 3.41 | 6.33 | 0.48 | 5.25 | 25.32 |

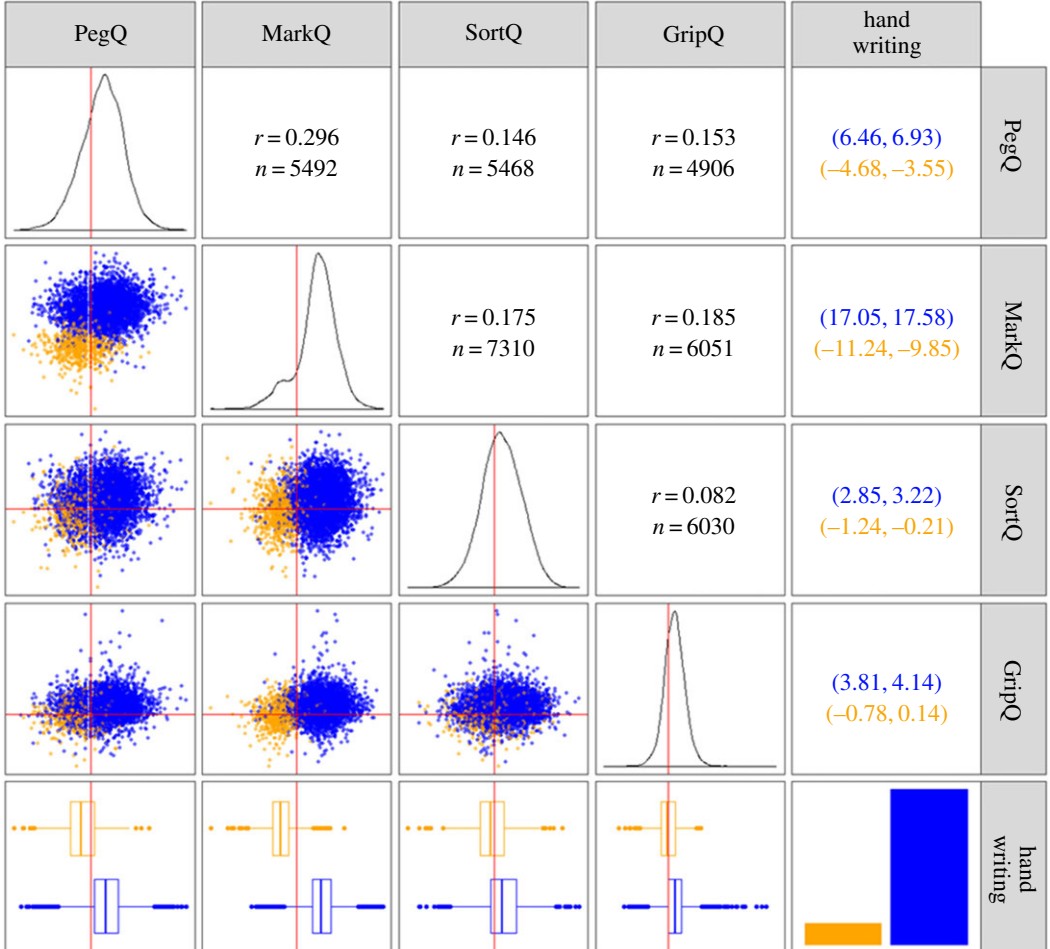

**Figure 1.** Correlation across laterality measures. The cells along the diagonal show the empirical distribution for each index and, in the last box, the bar-plot for the preferred hand for writing. The boxes on the left of the main diagonal show the bivariate distribution of the indexes colour coded for preferred hand for writing (left, orange; right, blue). For example, the first box on the second row of the matrix illustrates the PegQ scores on the x-axis and MarkQ on the y-axis. The bottom row shows the box-plots for each laterality index grouped by preferred hand for writing. The red lines are aligned along the zero for each index. The cells on the right of the diagonal show the Pearson correlation coefficients and the sample sizes from which they were calculated. The last column reports the confidence intervals for the box-plots shown in the bottom row, providing a measure of how the different indexes separate the right and left handers for preferred writing hand.

There was no age effect on the indexes (electronic supplementary material, figure S2) and no age difference between males and females (electronic supplementary material, table S1).

## 3.2. Correlation across measures

We assessed the joint distributions of the indexes both graphically and with Pearson correlation coefficients (figure 1). In general, the correlations between indexes were low, ranging from 0.08 (SortQ-GripQ) to 0.3 (MarQ-PegQ). MarkQ was the index that best predicted the preferred hand for writing. This effect was also reflected by the bimodal distribution of MarkQ which separates individuals with a right and left writing preference (see bottom row of figure 1). While not obviously bimodal, PegQ is the second-best index at separating left and right writing preference. SortQ and GripQ do not effectively separate the two groups. There was no substantial difference in this pattern when the analysis was conducted for male and females separately (electronic supplementary material, figure S3). Instead, analysis conducted separately in individuals who preferred to write with the right and left hand showed the correlations disappearing across the indexes (electronic supplementary material, figure S4). All indexes had a positive and negative mean when calculated for the right and left handers, respectively (table 3), but it was close to zero for SortQ and GripQ in the individuals who preferred writing with the left hand. Most right handers performed better with their right hand on all indexes. Only 3.6% right handers

**Table 3.** Laterality indexes in right and left handers.

| index | index distributions: right | | | | | | index distributions: left | | | | | |
| | N | mean | s.d. | skew | excess kurtosis | % negative scores | N | mean | s.d. | skew | excess kurtosis | % positive scores |
|---|---|---|---|---|---|---|---|---|---|---|---|---|
| PegQ | 6040 | 6.7 | 9.25 | −0.18 | 0.55 | 18.44 | 844 | −4.11 | 8.38 | −0.11 | 0.58 | 34.95 |
| MarkQ | 5622 | 17.32 | 10.13 | 0.05 | 0.92 | 3.61 | 783 | −10.54 | 9.87 | 0.01 | 1.73 | 13.41 |
| SortQ | 5598 | 3.04 | 7.01 | −0.07 | 0.13 | 29.96 | 783 | −0.72 | 7.31 | −0.02 | 0.7 | 48.28 |
| GripQ | 5020 | 3.97 | 6.07 | 0.66 | 4.73 | 21.51 | 713 | −0.32 | 6.23 | −0.18 | 1.01 | 48.53 |

**Table 4.** Principal component analysis.

|  | PC1 | PC2 | PC3 | PC4 |
|---|---|---|---|---|
| standard deviation | 1.244 | 0.962 | 0.914 | 0.831 |
| proportion of variance | 0.387 | 0.232 | 0.209 | 0.173 |
| cumulative proportion | 0.387 | 0.619 | 0.828 | 1.000 |
| loadings |  |  |  |  |
| PegQ | 0.552 | 0.009 | 0.579 | 0.601 |
| MarkQ | 0.593 | −0.016 | 0.234 | −0.770 |
| SortQ | 0.411 | 0.719 | −0.544 | 0.136 |
| GripQ | 0.419 | −0.695 | −0.561 | 0.166 |

performed better with their left hand on MarkQ. This fraction was higher in the other indexes reaching about 30% for SortQ. In comparison, a higher proportion of left handers performed better with their right hand. This ranged from 13.4% for MarkQ to over 48% for both SortQ and GripQ.

The structure amongst indexes was further explored with principal component analysis (PCA; table 4 and figure 2). The analysis was performed in 4569 individuals who had no missing data for the four indexes as well as for the preferred hand for writing. The first component, PC1, explaining 38.7% of the variation, gives broadly equal weight to each variable, suggesting that, even though the correlations between the variables are not strong, they are all measuring the same underlying trait and indeed this component shows discrimination between the left and right preferred hand for writing.

The remaining three components explain similar proportions of variance, and so may be somewhat arbitrarily ordered. This suggests that in addition to a general laterality trait, each index is capturing a different characteristic. The components can be interpreted as a contrast between SortQ and GripQ (PC2), a contrast between PegQ and MarkQ (PC4), and a contrast between a combined PegQ/MarkQ and a combined SortQ/GripQ (PC3). The analysis shows that PC3 does not greatly discriminate between left and right preference for writing hand (figure 2).

## 3.3. Sex effects

For each task, we estimated the effects of sex and hand preference on performance, as well as the interaction of these two factors (table 5). We used the best score regardless of which hand was used as a measure of performance (see electronic supplementary material, figure S5 to visualize the performance of the left versus right hand). An ANOVA (type II) showed that sex had an effect on performance on all tasks ($p < 0.0001$). Specifically, as a group, females outperformed males in dexterity on the pegboard (males are between 0.92 and 1.24 s slower), marking squares (males mark between 5.54 and 8.08 less squares) and sorting matches (males are between 1.24 and 1.89 s slower). Instead on strength, males have a grip between 0.85 and 1.31 kg stronger than females. We found no evidence that the preferred hand for writing had an effect on the performance in any of the tasks. There was no effect either for the interaction between hand preference and sex.

Although age had an effect on performance, especially for grip strength (pegboard: $r = −0.13$, $p < 0.0001$; marking squares: $r = 0.018$, $p = 0.13$; sorting matches: $r = −0.055$, $p < 0.0001$; grip strength: $r = 0.15$, $p < 0.0001$ electronic supplementary material, figure S6) there was no substantial age difference between males and females (electronic supplementary material, table S2) that could explain these sex effects.

Given these effects, and the higher frequency of left-hand preference in males compared to females (table 1), we assessed sex effects on the laterality indexes. Comparison of the means of the distributions consistently showed a shift towards the left and right for males and females, respectively (table 6). Of the principal components, only PC1, which captures a general laterality trait, shows strong differences between the sexes. The remaining components show little difference, suggesting that the indexes do not vary substantially according to sex.

To further assess sex effects on the indexes, we analysed the males/females ratios along the distributions of the indexes. The ratios consistently decreased from the negative to positive scores for all indexes (figure 3). This observation indicates an over-representation of males and females in left- and right-lateralized individuals, respectively. SortQ showed the most linear gradient with the most extreme values ranging from a male/female ratio of 1.43 in the most left-lateralized decile and of 0.77 in the most right-lateralized decile.

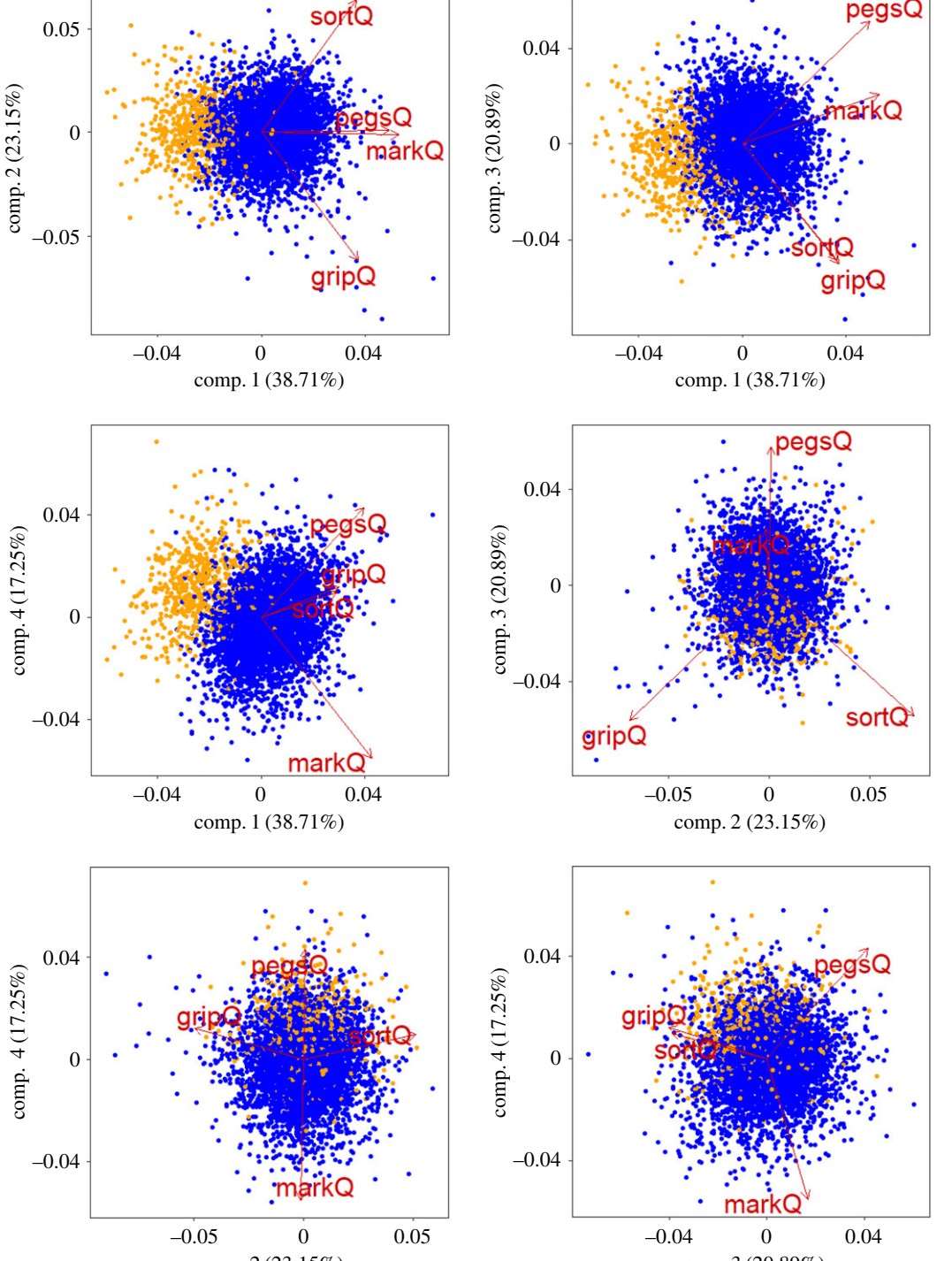

**Figure 2.** Biplots for all the PCA pairs. Each biplot visualizes the contribution of all indexes to a principal component pair. The length of the arrows illustrates the contribution to each principal component. The colours indicate the preferred hand for writing for each individual: orange, left and blue, right.

## 4. Discussion

We analysed different handedness-related measures testing specifically how they correlate with each other and how they are influenced by sex. We analysed four laterality indexes (PegQ, MarkQ, SortQ and MarkQ) and hand preference for writing in the ALSPAC dataset.

All indexes showed a majority of individuals performing better with the right hand (table 2), but presented only a moderate correlation with hand preference for writing (figure 2). MarkQ was the

**Table 5.** Effect of sex and hand on task performance preference.

| task (unit) | ANOVA *p*-value | | | estimated effects (95% confidence interval) | | | |
| --- | --- | --- | --- | --- | --- | --- | --- |
| | sex | hand for writing | interaction | base level: female, right hand | male | left hand | male and left-hand interaction |
| pegboard (seconds) | <0.0001 | 0.1485 | 0.365 | (21.15, 21.38) | (0.92, 1.24) | (−0.06, 0.64) | (−0.68, 0.25) |
| marking squares (marked squares) | <0.0001 | 0.7304 | 0.1306 | (83.64, 85.41) | (−8.08, −5.54) | (−3.99, 1.48) | (−0.84, 6.47) |
| sorting matches (seconds) | <0.0001 | 0.8465 | 0.1331 | (37.59, 38.05) | (1.24, 1.89) | (−0.26, 1.16) | (−1.67, 0.22) |
| grip strength (kilograms) | <0.0001 | 0.3852 | 0.1027 | (18.16, 18.47) | (0.85, 1.31) | (−0.94, 0.04) | (−0.11, 1.2) |

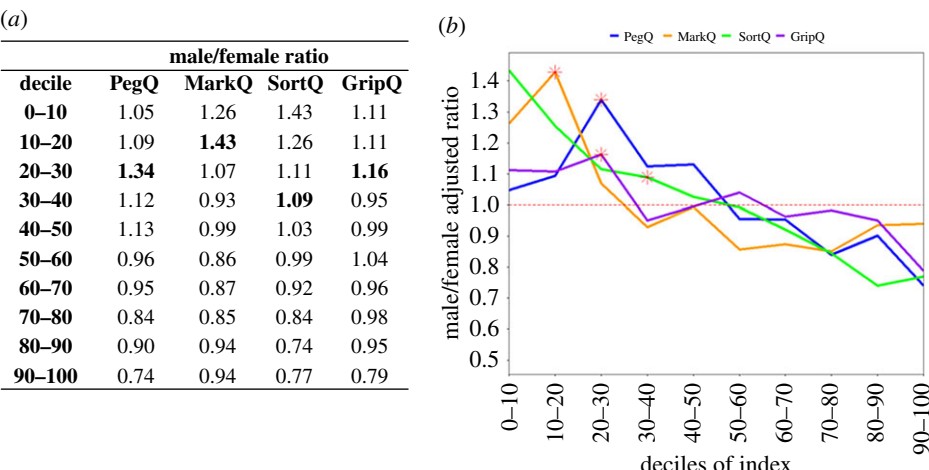

(a)

| decile | male/female ratio | | | |
|---|---|---|---|---|
| | PegG | MarkQ | SortQ | GripQ |
| **0–10** | 1.05 | 1.26 | 1.43 | 1.11 |
| **10–20** | 1.09 | **1.43** | 1.26 | 1.11 |
| **20–30** | **1.34** | 1.07 | 1.11 | **1.16** |
| **30–40** | 1.12 | 0.93 | **1.09** | 0.95 |
| **40–50** | 1.13 | 0.99 | 1.03 | 0.99 |
| **50–60** | 0.96 | 0.86 | 0.99 | 1.04 |
| **60–70** | 0.95 | 0.87 | 0.92 | 0.96 |
| **70–80** | 0.84 | 0.85 | 0.84 | 0.98 |
| **80–90** | 0.90 | 0.94 | 0.74 | 0.95 |
| **90–100** | 0.74 | 0.94 | 0.77 | 0.79 |

**Figure 3.** Males/females ratio across the distribution of the laterality indexes. The male/female ratios are shown as values (a) and visualized as graph (b). Each value has been corrected by the overall male/female ratio of data available for each index, i.e. PegQ = 1.0135 (=3465 M/3419F); MarkQ = 0.9657 (=3630 M/3759F); SortQ = 0.9701(=3627 M/3739F); GripQ = 0.964 (3271 M/3393F). The deciles including scores = 0 (i.e. equal performance with both hands on the corresponding task) are indicated in bold in (a) and with red asterisks in (b). The graph visualizes the consistent tendency for males and females to be more left- and right-lateralized, respectively.

**Table 6.** Comparison of index means in males and females. $\bar{x}$ = sample mean of the laterality index. $se = s/\sqrt{n}$, standard error of the sample mean. $\bar{x}_m - \bar{x}_f$ **95% C.I.** Confidence intervals for the difference of the means in males and females. C.I. that do not overlap zero, as in the case for all indexes, provide evidence that the distributions are different between the two sexes.

| | males | | | females | | | welch | |
|---|---|---|---|---|---|---|---|---|
| | $n_m$ | $\bar{x}_m$ | $se_m$ | $n_f$ | $\bar{x}_f$ | $se_f$ | t-test p-value | $\bar{x}_m - \bar{x}_f$ 95% C.I. |
| PegsQ | 3465 | 4.80 | 0.16 | 3419 | 5.95 | 0.17 | <0.0001 | (−1.61, −0.69) |
| MarkQ | 3630 | 13.09 | 0.23 | 3759 | 14.79 | 0.21 | <0.0001 | (−2.32, −1.08) |
| SortQ | 3627 | 1.88 | 0.12 | 3739 | 3.26 | 0.11 | <0.0001 | (−1.7, −1.05) |
| GripQ | 3271 | 3.17 | 0.11 | 3393 | 3.65 | 0.11 | 0.002 | (−0.78, −0.17) |
| PC1 | 2216 | −0.12 | 0.03 | 2353 | 0.11 | 0.03 | <0.0001 | (−0.3, −0.16) |
| PC2 | 2216 | −0.03 | 0.02 | 2353 | 0.03 | 0.02 | 0.0231 | (−0.12, −0.01) |
| PC3 | 2216 | 0.03 | 0.02 | 2353 | −0.03 | 0.02 | 0.051 | (0, 0.11) |
| PC4 | 2216 | −0.01 | 0.02 | 2353 | 0.01 | 0.02 | 0.6637 | (−0.06, 0.04) |

only bimodal index and it best separated the individuals with a left- and right-hand preference for writing. The marking squares test is based on the use of a pen, therefore it is possible that the proficiency acquired for the preferred writing hand might influence this laterality index. Marking squares data were collected at age 10, when the preferred hand for writing is well established and therefore one hand would be more skilled than the other at using a pen. Conversely, the other tasks are expected to be less influenced by this kind of training. For example, sorting matches and moving pegs are not daily activities on which an individual would be consistently exposed, contrary to holding a pen. After MarkQ, PegQ was the best measure at separating individuals for their preferred hand for writing. PegQ and MarkQ were the two indexes that showed the highest correlation, which was however quite modest ($r = 0.3$). Analysis in right- and left-handers separately shows that any correlation across indexes is completely lost within groups (electronic supplementary material, figure S4). These results are in line with those reported before [18] and suggest that the main factor influencing correlation across indexes is the direction rather than the degree of handedness. While all indexes had negative and positive means in the left- and right-handers respectively, the distributions

were not well separated. To different extents, all indexes showed that some individuals performed better with their non-preferred hand for writing at different tasks (table 3). This was particularly true for the left handers who were less consistent than right handers. Between 13.4% (for MarkQ) to over 48% (for SortQ and GripQ) of left-handers performed better with their right hand. The variance (s.d. in table 3) however was very similar for left- and right-handers for all indexes indicating that the difference in performance between the two hands was comparable in the two groups.

PCA confirmed that the four indexes both identify a general left/right dimension and are also all necessary to capture the structure of the data (table 4 and figure 3), suggesting no redundancy of any index. These data show that it remains a challenge to comprehensively assess degree of handedness and to reduce such assessment to a single measure. The relative hand difference, measured by each index, does not reflect a general laterality score but most likely depends on different skills required by the tasks. These performance tasks measure dexterity or strength using different skills. To start with, as we already discussed, marking squares is influenced by the training acquired in using pens or pencils with one hand preferentially. During the marking square an object (i.e. pencil) is continuously held and, in contrast, grasp and release are required by the pegboard and match sorting tasks, albeit with different levels of precision. In turn, the laterality indexes derived from a variety of skills might also be underpinned by different brain asymmetries. Some of the functions required by these tasks are lateralized and involved the two cerebral hemispheres in different ways. For example, motor control generally presents a left hemisphere dominance, while visuospatial attention tends to be controlled by right hemisphere [51]. Therefore, our results are in line with previous work [17,18,28], suggesting that handedness performance cannot be reduced to a single measure. Furthermore, while the direction of handedness can be reduced to left and right categories, the relative difference between the two hands cannot be categorized easily. The poor correlation between indexes indicates that the generation of a handedness factor score does not seem a reliable option. Our analysis also implies that comparisons across handedness studies that used different handedness measures are not a straightforward process. Accordingly, we strongly recommend to avoid referring to handedness as a generic measure or a universal concept, and encourage instead referring to the specific tasks used for handedness assessment. Our results also raise the question of whether different laterality phenotypes are underlined by shared biology. For example, the most recent GWAS ($N > 1.7$ million participants) for hand preference highlighted that the biological pathways implicated in handedness also contribute to disorders such as schizophrenia and appear to be mediated by mechanisms controlling the cell cytoskeleton [7,52]. Although associations were reported for different genes, previous genetic studies for continuous measures, and PegQ in particular, also suggested an overlap between the biology of handedness, neurodevelopmental disorders and cilia, a cellular structure controlled by cytoskeleton dynamics [6,53–56]. Therefore, current evidence supports some overlaps of the biological pathways contributing to different laterality measures, however, these hypotheses will require to be replicated more systematically.

As reported in previous literature [36], we found a higher frequency of males who preferred writing with the left hand compared to females (table 1). Also, consistent with previous literature, we found that females outperformed males in dexterity tasks (pegboard, marking squares and sorting matches) but not in strength (grip strength) tasks [37,40,41] (table 5). Some studies suggested that higher dexterity in females could be influenced by finger size [43], but these observations were limited to the pegboard task and failed to replicate in subsequent studies [44]. We did not have measures of finger sizes in our dataset and therefore could not test this effect directly. However, we report better performance in females on the marking squares task, which is unlikely to be affected by finger size. Therefore, our data are indeed suggestive of higher dexterity in females. By contrast, hand preference for writing did not have any effects on performance (table 5) either individually or when tested for interaction with sex.

Given the male/female differences in performance, we looked at sex effects on the laterality indexes. We found that males and females were more left- and right-lateralized, respectively, both in the dexterity- and strength-derived tasks (table 6). The males/females ratio was greater than 1 around the zero of all the distributions indicating a higher tendency for males to have similar scores for both hands or, in other words, to be weakly lateralized. To the best of our knowledge, the tendency for males and females to be over-represented at the opposite ends of the distributions of laterality indexes has not been reported before. Previous studies showed that females had bigger differences between the preferred and not preferred hand on dexterity performance [38–41]. However, if this was the case, we would expect an over-representation of females at both the left and right extremes. Instead, our analysis shows that females were over-represented only on the right side of the distribution for all indexes. We ruled out any potential age effects that could explain the sex effect on the indexes (electronic supplementary material, table S1).

In summary, we investigated different laterality measures in a large dataset. In agreement with previous literature, our results show a higher frequency of left-handedness in males and better dexterity performance in females. In addition, we report for the first time that males and females are more left- and right-lateralized on both dexterity and strength tasks. These data indicate the importance of factoring sex into any handedness and laterality study. The correlation across different indexes is weak, showing that different handedness tasks measure different handedness components and cannot be directly compared or combined under a general handedness label. Overall, this work provides a reference for the design and interpretation of handedness studies.

Ethics. Informed written consent was obtained from the parents after receiving a complete description of the study at the time of enrolment into the ALSPAC project, with the option for them or their children to withdraw at any time. Ethical approval for the study was obtained from the ALSPAC Law and Ethics Committee and the Local Research Ethics Committees.

Data accessibility. Data used for this submission will be made available on request to the Executive (alspac-exec@bristol.ac.uk). The ALSPAC data management plan (http://www.bristol.ac.uk/alspac/researchers/data-access/documents/alspac-data-management-plan.pdf) describes in detail the policy regarding data sharing, which is through a system of managed open access. All analyses scripts are available through Open Science Framework https://osf.io/4ysnk/.

Authors' contributions. S.P. conceived, designed and coordinated the study; C.B.C. analysed the data; S.P. and C.B.C. interpreted the data and drafted the manuscript; A.G.L. contributed important intellectual content to the analysis and the draft. All authors approved the final version to be published. S.P. will serve as guarantor for the contents of this paper.

Competing interests. The authors have no competing interests to declare.

Funding. S.P. is a Royal Society University Research Fellow. The UK Medical Research Council and Wellcome (grant ref: 102215/2/13/2) and the University of Bristol provide core support for ALSPAC. This publication is the work of the authors and A comprehensive list of grants funding is available on the ALSPAC website (http://www.bristol.ac.uk/alspac/external/documents/grant-acknowledgements.pdf).

Acknowledgements. We are extremely grateful to all the families who took part in this study, the midwives for their help in recruiting them, and the whole ALSPAC team, which includes interviewers, computer and laboratory technicians, clerical workers, research scientists, volunteers, managers, receptionists and nurses. We are grateful to Judith Schmitz for useful comment to the manuscript.

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
