## [Reviewer comments · Royal Society Open Science]

Review History

RSOS-191700.R0 (Original submission)

Review form: Reviewer 1

Is the manuscript scientifically sound in its present form?

Yes

Are the interpretations and conclusions justified by the results?

Yes

Is the language acceptable?

Yes

Do you have any ethical concerns with this paper?

No

Have you any concerns about statistical analyses in this paper?

No

Recommendation?

Major revision is needed (please make suggestions in comments)

Comments to the Author(s)

See review and annotated pdf for minor comments (Appendices A & B).

Review form: Reviewer 2

Is the manuscript scientifically sound in its present form?

Yes

Are the interpretations and conclusions justified by the results?

Yes

Is the language acceptable?

Yes

Do you have any ethical concerns with this paper?

No

Have you any concerns about statistical analyses in this paper?

No

Recommendation?

Accept with minor revision (please list in comments)

Comments to the Author(s)

Summary of Study

In this manuscript the authors evaluated the relationship between writing hand preference and its association with four other manual laterality dexterity indexes (pegboard task, marking squares & sorting matches, grip strength). Outcomes suggested there was a right hand performance bias at the population level – however each index had low correlation with other hand indexes. The authors interpreted the results to mean that different hand measures tap different contexts and are not interchangeable. An overarching theme across indexes was that at the population level, females demonstrated a higher rate of right hand bias than males (across indexes) and that males demonstrated a stronger tendency to be poorly lateralized compared with females.

General Comments:

The subject matter of the study is important and it is imperative that researchers carefully consider the type of test of handedness best suited for their research question because different tests will indeed tap different skills. I also fully agree with the author's statement recommending that researchers "avoid referring to handedness as a generic measure or a universal concept, and encourage instead to refer to the specific tasks used for handedness assessment". However, it is not surprising that these indexes did not all correlate with one another because they range from fine motor to gross motor to hand strength. It would have been interesting for the authors to consider these different types of skill that the tasks were eliciting and make some interpretations of the results based on what we know about brain organization for fine motor sequencing for both speech and hands.

There was the suggestion in the introduction that hand biases are tied to cognition, disorders and evolution, yet the discussion did not elaborate on any of these points. While the cause-effect relationship is debatable, behavioural markers are valuable and could drive additional research.

It was unclear if the longitudinal dataset could have included cognitive scores to help address how the different types of hand performance tasks associate with different cognitive skills. If there is the suggestion that the biases are linked with sex and disorders – as a reader I would like some framing of what the hand performance measures mean – E.g. how do these findings fit within an evolutionary or developmental framework – or supposition about the brain organization and cognition.

Abstract

There are some inconsistencies in the abstract that will confuse the reader. First it is suggested that males are more right biased but later in the manuscript it is reversed to suggest that females are more right biased. Second, the Grip Task is listed as a test of dexterity – where it is a gross motor test of strength. Finally, I am not certain that I would agree that performance tests are costly in comparison to other testing methods, but agree that they are time consuming.

Introduction Page 3 of 17:

Lines 7/8: The rightward prevalence of handedness is a feature specific to humans: This is vague because it does not reference any literature nor does it concede that apes show right handedness for tool use – although the prevalence/strength of handedness does not reach proportions found in humans (e.g. references).

Lines 9/10: This statement also needs a reference.

Lines 11/14: This statement needs some references – and suggests that the study might also consider the evolutionary perspective and/or links between hand dominances and cognition.

Lines 18/19: The supposition based on the reference that handedness is a developmental trait tied to language without a consideration that it has an evolutionary history bound to functional brain organisation other than language.

Lines 28/29: The EDI is also context specific – providing only tool using exemplars. Studies that consider hand actions across a variety of context find that dominances shift.

Lines 56/57: Is this the hypothesis? Maybe reframe as the research question? ‘E.g. One would assume that...?’

Introduction Page 4 of 17

Lines 3/4: It is agreed that performance tests can be time consuming, but are they considered expensive as a methodological approach in comparison to other experimental testing?

Lines 14/15: In the abstract it reads as if males are more right lateralised and females more left lateralised – but here it is the other way around.

What is the rationale for adding grip strength to a test of hand dominance for dexterity? Why would we expect correlation?

Lines 38/39: It would be useful if the authors could offer perspective on why the measures do not correlate – what different skills are we tapping – might they be governed by different and/or overlapping mechanisms? Do the measures correlate with cognition?

Materials and Methods page 5 of 17

Lines 27: capitalise 'Marking Square and Sorting Matches' is sometimes in capitals and sometimes not – please be consistent.

Results

It seems likely that the reason that the MarkQ and PegQ were most highly correlated with Hand Writing because they required the fine motor grip type as handwriting – whereas the SortQ was more gross motor and GripQ was about strength and not dexterity. This could be elaborated on in the discussion.

Decision letter (RSOS-191700.R0)

22-Nov-2019

Dear Miss Paracchini,

The editors assigned to your paper ("Laterality indexes are poorly correlated but consistently show the tendency of males and females to be more left- and right-lateralised, respectively") have now received comments from reviewers. We would like you to revise your paper in accordance with the referee and Associate Editor suggestions which can be found below (not including confidential reports to the Editor). Please note this decision does not guarantee eventual acceptance.

Please submit a copy of your revised paper before 15-Dec-2019. Please note that the revision deadline will expire at 00.00am on this date. If we do not hear from you within this time then it will be assumed that the paper has been withdrawn. In exceptional circumstances, extensions may be possible if agreed with the Editorial Office in advance. We do not allow multiple rounds of revision so we urge you to make every effort to fully address all of the comments at this stage. If deemed necessary by the Editors, your manuscript will be sent back to one or more of the original reviewers for assessment. If the original reviewers are not available, we may invite new reviewers.

- Data accessibility

If you wish to submit your supporting data or code to Dryad (<http://datadryad.org/>), or modify your current submission to dryad, please use the following link:
<http://datadryad.org/submit?journalID=RSOS&manu=RSOS-191700>

- Competing interests

- Authors' contributions

- Acknowledgements

- Funding statement

Kind regards,

Royal Society Open Science Editorial Office
Royal Society Open Science
openscience@royalsociety.org

on behalf of the Associate Editor, and Professor Essi Viding (Subject Editor)
openscience@royalsociety.org

Associate Editor's comments:

Two reviewers have provided commentary on your paper, and the general view appears to be the manuscript is heading in the right direction, but requires some work to get it over the line. Please ensure you respond fully to the reviewer comments in both a tracked-change version of the manuscript, and a point-by-point response detailing how you've addressed the reviewers' concerns.

Additionally, we note that, although only one author is listed on the electronic submission form, a number of co-authors are included in the manuscript. Please ensure you list all authors in both the form and the manuscript, and bear in mind that all authors conform to the requirements at <https://royalsociety.org/journals/ethics-policies/openness/>. If you need to make any changes to the authorship, this must be communicated to the editorial office, and ALL authors must agree the changes before they are made.

We look forward to receiving your revision.

Associate Editor: 2

Comments to the Author:

Thank you for this interesting manuscript. The large sample size and range of measures is a welcome contribution.

Comments to Author:

Reviewers' Comments to Author:

Reviewer: 1

Comments to the Author(s)

See review and annotated pdf for minor comments.

Reviewer: 2

Comments to the Author(s)

Summary of Study

In this manuscript the authors evaluated the relationship between writing hand preference and its association with four other manual laterality dexterity indexes (pegboard task, marking squares & sorting matches, grip strength). Outcomes suggested there was a right hand performance bias at the population level – however each index had low correlation with other hand indexes. The authors interpreted the results to mean that different hand measures tap different contexts and are not interchangeable. An overarching theme across indexes was that at the population level, females demonstrated a higher rate of right hand bias than males (across indexes) and that males demonstrated a stronger tendency to be poorly lateralized compared with females.

General Comments:

The subject matter of the study is important and it is imperative that researchers carefully consider the type of test of handedness best suited for their research question because different tests will indeed tap different skills. I also fully agree with the author's statement recommending that researchers "avoid referring to handedness as a generic measure or a universal concept, and encourage instead to refer to the specific tasks used for handedness assessment". However, it is not surprising that these indexes did not all correlate with one another because they range from fine motor to gross motor to hand strength. It would have been interesting for the authors to consider these different types of skill that the tasks were eliciting and make some interpretations

of the results based on what we know about brain organization for fine motor sequencing for both speech and hands.

There was the suggestion in the introduction that hand biases are tied to cognition, disorders and evolution, yet the discussion did not elaborate on any of these points. While the cause-effect relationship is debatable, behavioural markers are valuable and could drive additional research. It was unclear if the longitudinal dataset could have included cognitive scores to help address how the different types of hand performance tasks associate with different cognitive skills. If there is the suggestion that the biases are linked with sex and disorders – as a reader I would like some framing of what the hand performance measures mean – E.g. how do these findings fit within an evolutionary or developmental framework – or supposition about the brain organization and cognition.

Abstract

There are some inconsistencies in the abstract that will confuse the reader. First it is suggested that males are more right biased but later in the manuscript it is reversed to suggest that females are more right biased. Second, the Grip Task is listed as a test of dexterity – where it is a gross motor test of strength. Finally, I am not certain that I would agree that performance tests are costly in comparison to other testing methods, but agree that they are time consuming.

Introduction Page 3 of 17:

Lines 7/8: The rightward prevalence of handedness is a feature specific to humans: This is vague because it does not reference any literature nor does it concede that apes show right handedness for tool use – although the prevalence/strength of handedness does not reach proportions found in humans (e.g. references).

Lines 9/10: This statement also needs a reference.

Lines 11/14: This statement needs some references – and suggests that the study might also consider the evolutionary perspective and/or links between hand dominances and cognition.

Lines 18/19: The supposition based on the reference that handedness is a developmental trait tied to language without a consideration that it has an evolutionary history bound to functional brain organisation other than language.

Lines 28/29: The EDI is also context specific – providing only tool using exemplars. Studies that consider hand actions across a variety of context find that dominances shift.

Lines 56/57: Is this the hypothesis? Maybe reframe as the research question? 'E.g. One would assume that...?'

Introduction Page 4 of 17

Lines 3/4: It is agreed that performance tests can be time consuming, but are they considered expensive as a methodological approach in comparison to other experimental testing?

Lines 14/15: In the abstract it reads as if males are more right lateralised and females more left lateralised – but here it is the other way around.

What is the rationale for adding grip strength to a test of hand dominance for dexterity? Why would we expect correlation?

Lines 38/39: It would be useful if the authors could offer perspective on why the measures do not

correlate – what different skills are we tapping – might they be governed by different and/or overlapping mechanisms? Do the measures correlate with cognition?

Materials and Methods page 5 of 17

Lines 27: capitalise 'Marking Square and Sorting Matches' is sometimes in capitals and sometimes not – please be consistent.

Results

It seems likely that the reason that the MarkQ and PegQ were most highly correlated with Hand Writing because they required the fine motor grip type as handwriting – whereas the SortQ was more gross motor and GripQ was about strength and not dexterity. This could be elaborated on in the discussion.

Author's Response to Decision Letter for (RSOS-191700.R0)

See Appendix C.

RSOS-191700.R1 (Revision)

Review form: Reviewer 1

Is the manuscript scientifically sound in its present form?

Yes

Are the interpretations and conclusions justified by the results?

Yes

Is the language acceptable?

Yes

Do you have any ethical concerns with this paper?

No

Have you any concerns about statistical analyses in this paper?

No

Recommendation?

Accept as is

Comments to the Author(s)

mcuh improved. Well done chaps! Stay safe.

Review form: Reviewer 2

Is the manuscript scientifically sound in its present form?

Yes

Are the interpretations and conclusions justified by the results?

Yes

Is the language acceptable?

Yes

Do you have any ethical concerns with this paper?

No

Have you any concerns about statistical analyses in this paper?

No

Recommendation?

Accept as is

Comments to the Author(s)

Thank you for the opportunity to re-review this manuscript. The authors have gone to great lengths to address every point in turn that has been raised by the original review process. I appreciate the time and level of detail that has gone into this revision and think that the current submission is a greatly strengthened manuscript with clear rationale, categorisation of testing elements and measures and a thoughtful interpretation. The new analyses adds additional strength and a more comprehensive view of the handedness patterns. I am satisfied the the authors have appropriately addressed any points of weakness or ambiguity raised by the reviewers.

Decision letter (RSOS-191700.R1)

18-Mar-2020

Dear Dr Paracchini,

It is a pleasure to accept your manuscript entitled "Laterality indexes are poorly correlated but consistently show the tendency of males and females to be more left- and right-lateralised, respectively" in its current form for publication in Royal Society Open Science. The comments of the reviewer(s) who reviewed your manuscript are included at the foot of this letter. We hope that in these troubled times, this is a small bit of good news.

You can expect to receive a proof of your article in the near future. Please contact the editorial office (opencscience_proofs@royalsociety.org) and the production office (opencscience@royalsociety.org) to let us know if you are likely to be away from e-mail contact -- if

you are going to be away, please nominate a co-author (if available) to manage the proofing process, and ensure they are copied into your email to the journal.

on behalf of Essi Viding (Subject Editor)
openscience@royalsociety.org

Associate Editor Comments to Author:

Comments to the Author:

Thanks for taking such care in preparing your revision - the referees are now in favour of acceptance, congratulations!

Reviewer comments to Author:

Reviewer: 2

Comments to the Author(s)

Thank you for the opportunity to re-review this manuscript. The authors have gone to great lengths to address every point in turn that has been raised by the original review process. I appreciate the time and level of detail that has gone into this revision and think that the current submission is a greatly strengthened manuscript with clear rationale, categorisation of testing elements and measures and a thoughtful interpretation. The new analyses adds additional strength and a more comprehensive view of the handedness patterns. I am satisfied the the authors have appropriately addressed any points of weakness or ambiguity raised by the reviewers.

Reviewer: 1

Comments to the Author(s)

mcuh improved. Well done chaps! Stay safe.

Appendix A**ROYAL SOCIETY
OPEN SCIENCE**

Laterality indexes are poorly correlated but consistently show the tendency of males and females to be more left- and right-lateralised, respectively

Journal:	Royal Society Open Science
Manuscript ID	RSOS-191700
Article Type:	Research
Date Submitted by the Author:	26-Sep-2019
Complete List of Authors:	Paracchini, Silvia; University of St Andrews, School of Medicine
Subject:	neuroscience < BIOLOGY, behaviour < BIOLOGY
Keywords:	handedness, laterality, hand skills, ALSPAC, sex effect, human behaviour
Subject Category:	Psychology and cognitive neuroscience

Author-supplied statements

Relevant information will appear here if provided.

Ethics

Does your article include research that required ethical approval or permits?:

Yes

Statement (if applicable):

Informed written consent was obtained from the parents after receiving a complete description of the study at the time of enrolment into the ALSPAC project, with the option for them or their children to withdraw at any time. Ethical approval for the study was obtained from the ALSPAC Law and Ethics Committee and the Local Research Ethics Committees.

Data

It is a condition of publication that data, code and materials supporting your paper are made publicly available. Does your paper present new data?:

My paper has no data

Statement (if applicable):

This paper describes secondary analysis on existing data. **Data used for this submission will be made available on request to the Executive (alspac-exec@bristol.ac.uk).** The ALSPAC data management plan (<http://www.bristol.ac.uk/alspac/researchers/data-access/documents/alspac-data-management-plan.pdf>) describes in detail the policy regarding data sharing, which is through a system of managed open access.

All analysis scripts are available through Open Science Framework <https://osf.io/4ysnk/>.

Conflict of interest

I/We declare we have no competing interests

Statement (if applicable):

CUST_STATE_CONFLICT :No data available.

Authors' contributions

This paper has multiple authors and our individual contributions were as below

Statement (if applicable):

SP conceived the study; CBC analysed the data; SP and CBC drafted the manuscript; CBC, AGL and SP interpreted the results, finalised the manuscript and approved the final submission. SP will serve as guarantor for the contents of this paper.

Different laterality indexes are poorly correlated with one another but consistently show the tendency of males and females to be more left- and right- lateralised, respectively

Carlos Buenaventura Castillo^{1,2}, Andy Graeme Lynch^{1,2} and Silvia Paracchini^{1*}

¹School of Medicine, University of St Andrews, Scotland

²School of Mathematics and Statistics, University of St Andrews, Scotland

*correspondence to: sp58@st-andrews.ac.uk

Abstract

Handedness is assessed primarily as a binary trait on the basis of the preferred hand for writing. At population level, about 90% people prefer using the right. Handedness can also be assessed as a continuous trait with laterality indexes, but these are not time and cost effective, and are not routinely collected. Here, we assessed the relationship of writing hand preference with four laterality indexes derived from measures of dexterity (pegboard task, marking squares and sorting matches) and strength (grip strength) available in a range of N = 6664-8069 children from the ALSPAC cohort. Although all indexes identified a higher proportion of individuals performing better with their right hand, they showed low correlation with each other (0.08-0.3). Analysis of sex effects on the laterality indexes showed that males and females tend to be, on all measures, more right- and the left-lateralised, respectively. Males also had a higher tendency to be poorly lateralised. This study shows that different handedness measures tap into different dimensions of laterality and cannot be used interchangeably. The similar trends for males and females observed across indexes suggest that sex effects should be taken into account in handedness and laterality studies.

Keywords: handedness, laterality, hand skills, behaviour, sex effect, ALSPAC

Introduction

Worldwide, the vast majority of people (roughly 90%) prefer using the right hand for most tasks in contrast to a minority of about 10% who prefer the left hand (1, 2). The rightward prevalence of handedness is a feature specific to humans. Language, which is unique to humans is also lateralised with a strong dominance for language processing in the left hemisphere. These observations have led to a number of studies investigating the role of handedness and brain asymmetries both in the context of human evolution and in cognitive abilities. Handedness has been investigated for association with cognitive skills, personality traits and psychiatric disorders (3–5). However, the cause/effect relationship between handedness, brain asymmetries and disorders remains unexplained and debated (6).

Intuitively, handedness is a binary category and most studies rely on the assignment of participants to a left/right-hand status based on preferred writing hand. Although a binary classification has many advantages, such as convenience and cost-effectiveness, it is has been argued that the preferred hand for writing is not a sensitive measure and does not capture a more general handedness assessment (7, 8). The Edinburgh Handedness Inventory (EHI) provides a solution by deriving a laterality quotient through hand preference scores on a series of tasks. These include items, such as brushing teeth, which are not expected to be influenced by cultural pressures. In fact, it is well established that environmental factors might force the use of the right hand for writing in people who might prefer using the left hand (10). This phenomenon was mainly true for past generations, and still applies in some cultures (1, 11). Typically, handedness questionnaires are characterised by a J-shaped distribution, indicating that the majority of people present an overall right- or left- hand preference with a few individuals in between.

Instead, relative hand skill tests lead to quantitative and continuous indexes of handedness. The indexes are derived by comparing performance of the two hands in carrying out skilled tasks. Most typically, these tests measure speed and dexterity, such as the Annett pegboard (12), marking squares and sorting matches tasks (13). The grip-strength test instead measures differences in manual strength (14). Test-retest correlations show that these tasks are sufficiently reliable (15, 16). Handedness indexes are derived by comparing the scores of the right (R) and left (L) hand (see Methods) and lead to continuous distributions with positive means indicating a higher number of individuals who perform better with the right hand.

The higher frequency of better right-hand performance suggests there will a rough correlation across indexes. A difficulty in demonstrating this assumption is the availability of adequate datasets. While hand preference data are easy to collect in extremely large samples through self-reported

1
2
3 questionnaire (17), performance tests are expensive and time-consuming. Therefore, collecting
4 performance data in large samples is challenging, making it difficult to compare and evaluate
5 correlations across tests. Multiple tasks available for the same participants are usually limited to a
6 couple of measures as in the case of The National Child Development Study (NCDS) (18, 19) for which
7 marking square and sorting matches tests were collected in over 6000 children (13). Only a few studies
8 have examined the relationship across tests (15, 20). 
14 Hand preference has been extensively analysed for the effect of sex, and males consistently show a
15 higher prevalence of left-handedness (21). Although sex effects have been investigated in
16 performance tasks, less focus has been given to laterality indexes. In general, females have been
17 reported to perform faster with their favourite hand on the dexterity tests (22–25). Some of these
18 effects have been suggested to be the results of smaller finger sizes in females compared to males, at
19 least for the pegboard test (22, 26, 27) but this interpretation is not consistently supported (28).
20 Instead, males tend to have higher scores than females in strength tests (29). 
26 By taking advantage of the Avon Longitudinal Study of Parents and Children (ALSPAC), we aimed to
27 address some of these issues around handedness measurements. We analysed different handedness
28 indexes derived from the pegboard, marking squares, sorting matches and grip strength tasks in a
29 range of 6664-8069 children. The indexes showed a higher proportion of children performing better
30 with their right hand but were poorly correlated with one another. Consistently with previous
31 literature, we found a higher frequency of left-handedness in males. Sex effects on the handedness
32 indexes showed that male and females tend to be more left- and right-lateralized, respectively. These
33 data show that different handedness measures are not interchangeable but are similarly influenced
34 by sex.

44 **Material and Methods Samples**

47 *The ALSPAC cohort.*

49 ALSPAC is a longitudinal cohort representing the general population living in the Bristol area. The
50 ALSPAC cohort consists of over 15,000 children from the southwest of England that had expected
51 dates of delivery between 1st April 1991 and 31st December 1992 (30, 31). From age 7, all children
52 were invited annually for assessments on a wide range of physical, behavioural, and
53 neuropsychological traits, including cognitive (reading and mathematics related) measures.
54 Attendance to the annual assessment determined the availability of data for the measures used in this
55 study. Informed written consent was obtained from the parents after receiving a complete description
56
57
58
59
60

1
2
3 of the study at the time of enrolment into the ALSPAC project, with the option for them or their
4 children to withdraw at any time. Ethical approval for the study was obtained from the ALSPAC Law
5 and Ethics Committee and the Local Research Ethics Committees.
6
7
8
9

10 *Phenotypes*

11 Self-reported hand preference for writing was collected at age seven.

12
13
14
15 The pegboard task was conducted as part of the movement assessment battery for children
16 (Movement ABC; (32)). The child had to insert twelve pegs, one at a time, into a peg board, holding
17 the board with one hand and inserting the pegs with the other, as quickly as possible. The task was
18 carried out with the preferred and the non-preferred hand, after it had been described and
19 demonstrated by the tester, and after a practice with each hand. The score corresponded to the time
20 taken to complete the task with each hand.
21
22
23
24
25

26 The marking **Square and Sorting Matches tasks** are a repetition of those used in the NCDS (33) and
27 were collected at age 10. Both tasks were first demonstrated by the tester and then the child had a
28 practice. Scores were then collected from two measurements for each hand.
29
30

31 In the marking square task, the child is asked to make a short dash with a pencil on a piece of paper
32 which has a grid made of rows of 20 squares. They are asked to start at the top left-hand side of the
33 squared paper and move across it. When the first line is completed the child should move on the left
34 side of the next row. The score corresponded to the number of squares that could be marked in 60
35 seconds and it was derived from the mean score of the two trials.
36
37
38
39
40

41 In the sorting matches task, the child is asked to move one match at a time across two boxes, one
42 full and one empty using one hand only. The score corresponded to the time taken to transfer all
43 the matches from one box to the other.
44
45
46
47
48

49 Grip strength was assessed with a Jamar hand dynamometer at age 11. After one demonstration from
50 the tester, the child was given the opportunity to practice. The child was encouraged to squeeze the
51 apparatus as long and as strongly as possible. The measurements were taken alternating hands for
52 three times and starting with the right hand. The higher the reading (measured in kilograms), the
53 stronger the grip. The mean from the three measurements for each hand was used.
54
55
56
57
58
59
60

Please note that the ALSPAC website contains details of all the data that is available through a fully searchable data dictionary and variable search tool" and reference the following webpage <http://www.bristol.ac.uk/alspac/researchers/our-data/>

When scores for performance tasks were recorded for the "dominant/non-dominant" hand, we used information about the preferred hand for writing to define the scores for the left and right hand. When multiple trials were available a mean score across the trials was used. For all four performance tasks, we derived laterality indexes, namely PegQ, MarkQ, SOrtQ and GripQ, based on previous literature so that positive and negative scores corresponded to a better performance with the right and left hand, respectively(34, 35).

Data analysis

All analyses and data visualization were performed using the tidyverse and rlang packages R Studio v.3.5.1 (36). All analyses scripts are available through Open Science Framework <https://osf.io/4ysnk/>.

Data availability

Data used for this submission will be made available on request to the Executive (alspac-exec@bristol.ac.uk). The ALSPAC data management plan (<http://www.bristol.ac.uk/alspac/researchers/data-access/documents/alspac-data-management-plan.pdf>) describes in detail the policy regarding data sharing, which is through a system of managed open access.

RESULTS

Handedness indexes

Analysis was conducted using data from the ALSPAC cohort. We assessed the frequency of hand preference for writing recorded as a self-reported measure when children were 7 years old (N = 8069, Table 1). In total, 977 (12.1%) were left-handed. Consistently with previous literature (21), there was a higher proportion of left-handed males (13.6%) than females (10.5%). This measure did not identify any ambidextrous individuals.

Table 1: Distribution of hand preference by gender

	Right	Left	Total
Male	3,530 (86.4%)	558 (13.6%)	4,088
Female	3,562 (89.5%)	419 (10.5%)	3,981
Total	7,092 (87.9%)	977 (12.1%)	8,069

Four laterality indexes (PegQ, SortQ, MarkQ and GripQ) were derived from manual tasks that measure the performance with both the right (R) and left (L) hand. For all indexes a positive value indicates a better performance with the right hand (Table 2).

MarkQ presents a bimodal distribution, consistent with **previous reports (13)**, while the others are unimodal and well-approximated by a normal distribution (Table 2, Supplementary Fig S1). All indexes are leptokurtic and have a positive mean, indicating that the majority of individuals performed better with the right hand. PegQ and SortQ exhibit discrete behaviour near zero as expected from their construction (See Supplementary Figure S1). **MarkQ presented 13% negative scores**, similar to the frequency of individuals who preferred writing with the left hand. The other indexes had a larger proportion of negative scores (24%-32%; Table 2).

Most data were collected within a period of roughly three months around the target age for each test but the actual age range spanned across two years (Table 2; Supplementary Figure S2). There was no age effect on the indexes (Supplementary Figure S2) and no age difference between males and females (Supplementary Table S1).

**Table 2. Laterality indexes**

Index	Formula	N	Age (months)		N trials/ hand	Index distribution				
			Mean	SD		Mean	SD	Skew	Excess Kurtosis	% negative scores
PegQ	$100*(L-R)/(L+R)$	6884	92.3	3.9	1	5.37	9.81	-0.19	0.35	24.16
MarkQ	$100*(R-L)/(L+R)$	7389	130.1	3.2	2	13.95	13.59	-0.65	0.96	13.51
SortQ	$100*(L-R)/(L+R)$	7366	130.1	3.2	2	2.58	7.13	-0.07	0.18	32.65
GripQ	$100*(R-L)/(L+R)$	6664	143.4	2.9	3	3.41	6.33	0.48	5.25	25.32

Age was reported in days (as in Supplementary Table S1), but presented here as months for easier interpretation. See Supplementary Figure S1 for the distribution of the indexes and Supplementary Figure S2 for age effects on the indexes

Correlation across measures

We assessed the joint distributions of the indexes graphically, and with the Pearson correlation coefficient (Figure 1). In general, the correlations between indexes were low, ranging from 0.08 (SortQ-GripQ) to 0.3 (MarQ-PegQ). MarkQ was the index that best predicted the preferred hand for writing. This effect was also reflected by the bimodal distribution of MarkQ which separates individuals with a right and left writing preference. While not obviously bimodal, PegQ is the second-best index at separating left and right writing preferences. SortQ and GripQ do not effectively separate the populations. There was not substantial difference in this pattern when the analysis was conducted for male and females separately (Supplementary Figure S3).

Figure 1. Correlation across la laterality measures. The cells along the diagonal show the empirical distribution for each index and, in the last box, the bar-plot for the preferred hand for writing. The boxes on the left of the main diagonal show the bivariate distribution of the indexes colour coded for preferred hand for writing (left = orange; right = blue). For example, the first box on the second row of the matrix illustrates the PegQ scores on the x-axis and MarkQ on the y-axis. The bottom row shows the box-plots for each laterality index grouped by preferred hand for writing. The red lines are aligned along the zero for each index. The cells on the right of the diagonal show the correlation coefficients and the sample sizes from which they were calculated. The last column reports the confidence intervals for the box-plots shown in the bottom row, providing a measure of how the different indexes separate the right and left handers for preferred writing hand.

The structure amongst indexes was further explored with principal component analysis (PCA; Table 3 and Figure 2). The analysis was performed on 4,569 individuals who had no missing data for the four indexes as well as for the preferred hand for writing. The first component, PC1, explaining 38.7% of the variation, gives broadly equal weight to each variable, suggesting that, even though the correlations between the variables are not strong, they are all measuring the same underlying trait (handedness) and indeed this component shows discrimination between the left and right preferred hand for writing.

The remaining three components explain similar proportions of variance, and so may be somewhat arbitrarily ordered. This suggests that in addition to the common trait, each index is capturing a different characteristic. The components can be interpreted as a contrast between SortQ and GripQ (PC2), a contrast between PegQ and MarkQ (PC4), and a contrast between a combined PegQ/MarkQ and a combined SortQ/GripQ (PC3). Perhaps unexpectedly, given the results shown in Figure 1, PC3 does not greatly discriminate between left and right preference for writing hand (Figure 2).

Table 3. Principal Component Analysis

	PC1	PC2	PC3	PC4	
Standard deviation	1.244	0.962	0.914	0.831	
Proportion of Variance	0.387	0.232	0.209	0.173	
Cumulative Proportion	0.387	0.619	0.828	1.000	
Loadings	PegQ	0.552	0.009	0.579	0.601
	MarkQ	0.593	-0.016	0.234	-0.770
	SortQ	0.411	0.719	-0.544	0.136
	GripQ	0.419	-0.695	-0.561	0.166

Figure 2. Biplots for all the PCA pairs. Each biplot visualises the contribution of all indexes to a principal component pair. The length of the arrows illustrates the variable contribution to each principal component. The colours indicate the preferred hand for writing for each individual: orange = left and blue = right.

Sex effects

For each task, we first analysed gender effects on performance regardless of laterality, using a second independent two-sample t-test (Table 4). We used the best score regardless of which hand was used as a measure of performance (See supplementary Figure S4 to visualise the performance of the left versus right hand). We found significant differences between males and females in all the tasks ($p < 0.0001$). Females outperformed males in the pegboard, marking squares and sorting matches tasks, whereas males performed better in the grip strength task. Although age had an effect on performance, especially for grip strength (Pegboard: $r = -0.13$, $p < 0.0001$; Marking squares: $r = 0.018$, $p = 0.13$; Sorting matches: $r = -0.055$, $p < 0.0001$; Grip strength: $r = 0.15$, $p < 0.0001$ supplementary Figure S5) there was no age difference between males and females (Supplementary Table 1) that could explain these sex effects.

Table 4 Gender differences in performance of the best hand

Task (unit)	Males			Females			Welch t-test P-val	$\bar{x}_m - \bar{x}_f$ 95% C.I.
	n_m	\bar{x}_m	se_m	n_f	\bar{x}_f	se_m		
Pegboard (seconds)	3,465	22.35	0.06	3,419	21.29	0.05	<0.0001	(0.91, 1.21)
Marking Squares (marked squares)	3,630	77.62	0.39	3,759	84.07	0.41	<0.0001	(-7.56, -5.34)
Sorting matches (seconds)	3,627	39.35	0.11	3,739	37.84	0.10	<0.0001	(1.22, 1.8)
Grip strength (kilograms)	3,271	19.43	0.07	3,393	18.30	0.07	<0.0001	(0.94, 1.34)

\bar{x} = sample mean of the best hand performance; $\frac{s_x}{\sqrt{n}}$ = standard deviation of the sample mean

Better performance is represented by a lower score for pegboard and sorting matches, and a higher score for marking squares and grip strength.

$\bar{x}_m - \bar{x}_f$ 95% C.I. Confidence intervals for the difference of the means in males and females.

Given these differences, and the higher frequency of left-hand preference in males compared to females (Table 1), we assessed sex effects on the laterality indexes. Comparison of the means of the distributions consistently showed a shift towards the left and right for males and females, respectively (Table 5). Of the principal components, only PC1, which captures a general laterality trait, shows strong differences between the sexes. The remaining components show little difference, suggesting that the indexes do not vary substantially according to sex.

Table 5 Comparison of index means in males and females

	n_m	Males \bar{x}_m	se_m	n_f	Females \bar{x}_f	se_f	Welch t-test P-val	$\bar{x}_m - \bar{x}_f$ 95% C.I.
PegsQ	3,465	4.80	0.16	3,419	5.95	0.17	<0.0001	(-1.61, -0.69)
MarkQ	3,630	13.09	0.23	3,759	14.79	0.21	<0.0001	(-2.32, -1.08)
SortQ	3,627	1.88	0.12	3,739	3.26	0.11	<0.0001	(-1.7, -1.05)
GripQ	3,271	3.17	0.11	3,393	3.65	0.11	0.002	(-0.78, -0.17)
PC1	2,216	-0.12	0.03	2,353	0.11	0.03	<0.0001	(-0.3, -0.16)
PC2	2,216	-0.03	0.02	2,353	0.03	0.02	0.0231	(-0.12, -0.01)
PC3	2,216	0.03	0.02	2,353	-0.03	0.02	0.051	(0, 0.11)
PC4	2,216	-0.01	0.02	2,353	0.01	0.02	0.6637	(-0.06, 0.04)

\bar{x} = sample mean of the laterality index; $\frac{s}{\sqrt{n}}$ = standard deviation of the sample mean

$se = \frac{s}{\sqrt{n}}$, standard error

$\bar{x}_m - \bar{x}_f$ **95% C.I.** Confidence intervals for the difference of the means in males and females. C.I. that do not overlap zero, as in the case for all indexes, provide evidence that the distributions are different between the two sexes.

To further assess sex effects on the indexes, we analysed the males/females ratios along the distributions. The ratios consistently decreased from the negative to positive scores for all indexes (Figure 3). This observation indicates an over-representation of males and females in left- and right-lateralised individuals, respectively. SortQ showed the most linear gradient with the most extreme values ranging from a male/female ratio of 1.43 in the most left-lateralized decile and of 0.77 in the most right-lateralized decile.

Figure 3. Males/females ratio across the distribution of the laterality indexes. The male/female ratio are shown as values (A) and visualised as graph (B). Each value has been corrected for the overall male/female ratio of data available for each index, i.e. PegQ = 1.0135 (=3465M/3419F); MarkQ = 0.9657 (=3630M/3759F); SortQ = 0.9701(=3627M/3739F); GripQ = 0.964 (3271M/3393F). The deciles including scores = 0 (i.e. ambidextrous on the corresponding task) are indicated in bold in (A) and with red asterisks in (B). The graph visualises the consistent tendency for males and females to be more left- and right-lateralised, respectively.

DISCUSSION

We analysed different handedness-related measures testing specifically how they correlate with each other and how they are influenced by sex. We analysed four laterality indexes (PegQ, MarkQ, SortQ and MarkQ) and hand preference for writing in the ALSPAC dataset.

All indexes showed a majority of individuals performing better with the right hand (Table 2), but presented only a moderate correlation with hand preference for writing (Figure 2). MarkQ was the only bimodal index and best separated the individuals with a left- and right- hand preference for writing. The marking squares test is based on the use of a pen, therefore it is possible that the proficiency acquired for the preferred writing hand might influence this laterality index. Marking squares data were collected at age 10, when the preferred hand for writing is well established and therefore one hand would be more skilled than the other at using a pen. Conversely, the other tasks are expected to be less influenced by this kind of training. For example, sorting matches and moving pegs are not daily activities on which an individual would develop consistent exposure as opposed to holding a pen. After MarkQ, PegQ was the best measure at separating individuals for their preferred hand for writing. PegQ and MarkQ were the two indexes showing the highest correlation, which was however quite modest ($r = 0.3$).

Principal component analysis confirmed that the four indexes are necessary to capture the structure of the data (Table 3 and Figure 3), suggesting no redundancy of indexes. These data show that it remains a challenge to comprehensively assess handedness and to reduce such assessment in a single measure. For example, the poor correlation between indexes indicates that the generation of an handedness factor score does not seem a reliable option. Our analysis also implies that comparisons across handedness studies that used different handedness measures are not a straightforward process. In light of our current results, we strongly recommend to avoid referring to handedness as a generic measure or a universal concept, and encourage instead to refer to the specific tasks used for handedness assessment. Our results also raise the question of whether different laterality phenotypes are underlined by shared biology. For example, the most recent GWAS for hand preference highlighted that the biological pathways implicated in handedness also contribute to disorders such as schizophrenia and appear to be mediated by mechanisms controlling the cell cytoskeleton (4). Although associations were reported for different genes, previous genetic studies for continuous measures, and PegQ in particular, also suggested an overlap between the biology handedness and neurodevelopmental disorders and a role of cilia biology, a process closely link with cytoskeleton dynamics (3, 37–40).

1
2
3 As reported in previous literature (21), we found a higher frequency of males who preferred writing
4 with the left hand compared to females (Table 1). Also consistently with previous literature, we found
5 that females outperformed males in dexterity tasks (pegboard, marking squares and sorting matches)
6 but not strength (grip strength) tasks (24, 25, 29) (Table 4). Some studies suggested that higher
7 dexterity in females could be influenced by finger size (27) but these observations were limited to the
8 pegboard task and failed to replicate in subsequent studies (28). We did not have measures of finger
9 sizes in our dataset and therefore could not test this effect directly. However, we report better
10 performance in females on the marking squares task as well, which is unlikely to be affected by finger
11 size. Therefore, our data are indeed suggestive of higher dexterity in females.

12
13
14
15
16
17
18
19 Given the male/female differences in performance, we looked at sex effects on the laterality indexes.

20
21 We found that males and females were more left- and right-lateralised, respectively, both in the
22 dexterity- and strength-derived tasks (Table 5). The males/females ratio was > 1 around the zero of
23 the all distributions indicating a higher tendency for males to have similar scores for both hands or, in
24 other words, to be poorly lateralized. To the best of our knowledge, the tendency for males and
25 females to be over-represented at the opposite ends of the distributions of laterality indexes has not
26 been reported before. Previous studies, showed that females had bigger differences between the
27 preferred and not preferred hand on dexterity performance (22–25). However, if this was the case,
28 we would expect an over-representations of females at both the left and right extremes. Instead, our
29 analysis shows that females were over-represented only on the right side of the distribution for all
30 indexes. While the majority of individuals had the similar age when performing the different tasks, the
31 age range varied around a window of about two years. We ruled out any potential age effects that
32 could explain the sex effect on the indexes (Supplementary Table S1).

33
34
35
36
37
38
39
40
41 In summary, we investigated different laterality measures in a large dataset. In agreement with
42 previous literature, our results show a higher frequency of left-handedness in males and better
43 dexterity performance in females. In addition, we report for the first time that males and females are
44 more left- and right-lateralised on both dexterity and strength tasks. These data indicate the
45 importance of factoring sex into any handedness and laterality study. The correlation across different
46 indexes is weak, showing that different handedness tasks measure different handedness components
47 and cannot be directly compared or combined and interpreted under a general handedness label.
48 Overall, this work provides a reference for the design and interpretation of handedness studies
49 especially when for the use of different measures.
50
51
52
53
54
55
56
57
58
59
60

Acknowledgements

SP is a Royal Society University Research Fellow. The UK Medical Research Council and Wellcome (Grant ref: 102215/2/13/2) and the University of Bristol provide core support for ALSPAC. This publication is the work of the authors and Silvia Paracchini will serve as guarantor for the contents of this paper. A comprehensive list of grants funding is available on the ALSPAC website (<http://www.bristol.ac.uk/alspac/external/documents/grant-acknowledgements.pdf>). We are extremely grateful to all the families who took part in this study, the midwives for their help in recruiting them, and the whole ALSPAC team, which includes interviewers, computer and laboratory technicians, clerical workers, research scientists, volunteers, managers, receptionists and nurses. We are grateful to Judith Schmitz for useful comment to the manuscript.

References

1. Papadatou-Pastou M, Martin M, Munafo M, Ntolka E, Ocklenburg S, Paracchini S (2019): The prevalence of left-handedness: Five meta-analyses of 200 studies totaling 2,396,170 individuals. . doi: 10.31234/OSF.IO/5GJAC.
2. McManus IC (2002): *Right hand, left hand*. London, UK: Phoenix.
3. Brandler WM, Paracchini S (2014): The genetic relationship between handedness and neurodevelopmental disorders. *Trends Mol Med*, 2013/11/28. 20: 83–90.
4. Wiberg A, Ng M, Al Omran Y, Alfaro-Almagro F, McCarthy P, Marchini J, *et al.* (2019): Handedness, language areas and neuropsychiatric diseases: insights from brain imaging and genetics. *Brain*. . doi: 10.1093/brain/awz257.
5. Paracchini S, Diaz R, Stein J (2016): Advances in Dyslexia Genetics—New Insights Into the Role of Brain Asymmetries. *Adv Genet*. 96: 53–97.
6. Bishop DVM (2013): Cerebral asymmetry and language development: cause, correlate, or consequence? *Science (80-)*, 2013/06/15. 340: 1230531.
7. Paracchini S, Scerri T (2017): Genetics of human handedness and laterality. In: Rogers L, Vallortigara G, editors. *Lateralized brain Funct*. Springer, pp 523–552.
8. Scharoun SM, Bryden PJ (2014): Hand preference, performance abilities, and hand selection in children. *Front Psychol*. 5: 82.
9. Oldfield RC (1971): The assessment and analysis of handedness: the Edinburgh inventory. *Neuropsychologia*, 1971/03/01. 9: 97–113.
10. Porac C, Coren S, Searleman A (1986): Environmental factors in hand preference formation: Evidence from attempts to switch the preferred hand. *Behav Genet*. 16: 251–261.
11. de Kovel CGF, Carrión-Castillo A, Francks C (2019): A large-scale population study of early life factors influencing left-handedness. *Sci Rep*. 9: 584.
12. Annett M (1972): The distribution of manual asymmetry. *Br J Psychol*, 1972/08/01. 63: 343–358.
13. McManus IC (1985): Right- and left-hand skill: failure of the right shift model. *Br J Psychol*. 76 (Pt 1): 1–34.
14. Provins KA, Magliaro J (1993): The Measurement of Handedness by Preference and Performance Tests. *Brain Cogn*. 22: 171–181.
15. McManus IC, Van Horn JD, Bryden PJ (2016): The Tapley and Bryden test of performance differences between the hands: The original data, newer data, and the relation to pegboard and other tasks. *Laterality Asymmetries Body, Brain Cogn*. 21: 371–396.

16. Annett M, Hudson PTW, Turner A (1974): The reliability of differences between the hands in motor skill. *Neuropsychologia*. 12: 527–531.
17. Kovel CGF de, Francks C (2018): The molecular genetics of hand preference revisited. *bioRxiv*. 447177.
18. Davie R, Butler N, Goldstein H (1972): *From birth to seven: the second report of the national child development study. (1958 Cohort)*. London Longmans. .
19. Power C, Elliott J (2006): Cohort profile: 1958 British birth cohort (National Child Development Study). *Int J Epidemiol*. 35: 34–41.
20. Annett M (1992): Five Tests of Hand Skill. *Cortex*. 28: 583–600.
21. Papadatou-Pastou M, Martin M, Munafò MR, Jones G V. (2008): Sex differences in left-handedness: A meta-analysis of 144 studies. *Psychol Bull*. 134: 677–699.
22. Bryden PJ, Roy EA (2005): A new method of administering the Grooved Pegboard Test: Performance as a function of handedness and sex. *Brain Cogn*. 58: 258–268.
23. Bornstein RA (1985): Normative data on selected neuropsychological measures from a nonclinical sample. *J Clin Psychol*. 41: 651–659.
24. Thompson LL, Heaton RK, Matthews CG, Grant I (1987): Comparison of preferred and nonpreferred hand performance on four neuropsychological motor tasks. *Clin Neuropsychol*. 1: 324–334.
25. Schmidt SL, Oliveira RM, Rocha FR, Abreu-Villaca Y (2000): Influences of Handedness and Gender on the Grooved Pegboard Test. *Brain Cogn*. 44: 445–454.
26. Peters M, Servos P, Day R (1990): Marked sex differences on a fine motor skill task disappear when finger size is used as covariate. *J Appl Psychol*. 75: 87–90.
27. Peters M, Campagnaro P (1996): Do women really excel over men in manual dexterity? *J Exp Psychol Hum Percept Perform*. 22: 1107–1112.
28. Nicholson KG, Kimura D (1996): Sex Differences for Speech and Manual Skill. *Percept Mot Skills*. 82: 3–13.
29. Sella GE (2001): The hand grip: gender, dominance and age considerations. *Eura Medicophys*. 37: 161–170.
30. Boyd A, Golding J, Macleod J, Lawlor DA, Fraser A, Henderson J, *et al.* (2013): Cohort Profile: the 'children of the 90s'--the index offspring of the Avon Longitudinal Study of Parents and Children. *Int J Epidemiol*, 2012/04/18. 42: 111–127.
31. Fraser A, Macdonald-Wallis C, Tilling K, Boyd A, Golding J, Davey Smith G, *et al.* (2013): Cohort Profile: The Avon Longitudinal Study of Parents and Children: ALSPAC mothers cohort. *Int J Epidemiol*. 42: 97–110.
32. Henderson SE, Sugden DA (1992): *Movement Assessment Battery for Children manual*. Sidcup: The Psychological Corporation.
33. Leask SJ, Crow TJ (2001): Word acquisition reflects lateralization of hand skill. *Trends Cogn Sci*. 5: 513–516.
34. Annett M (1985): *Left, Right, Hand and Brain: The right Shift theory*. London: Psychology Press.
35. McManus IC (1985): Handedness, language dominance and aphasia: a genetic model. *Psychol Med Monogr Suppl*, 1985/01/01. 8: 1–40.
36. RStudioTeam (2015): *RStudio: Integrated Development for R*. Boston, MA: RStudio, Inc.
37. Scerri TS, Brandler WM, Paracchini S, Morris AP, Ring SM, Richardson AJ, *et al.* (2011): PCSK6 is associated with handedness in individuals with dyslexia. *Hum Mol Genet*, 2010/11/06. 20: 608–614.
38. Brandler WM, Morris AP, Evans DM, Scerri TS, Kemp JP, Timpson NJ, *et al.* (2013): Common variants in left/right asymmetry genes and pathways are associated with relative hand skill. *PLoS Genet*, 2013/09/27. 9: e1003751.
39. Francks C, Maegawa S, Lauren J, Abrahams BS, Velayos-Baeza A, Medland SE, *et al.* (2007): LRRTM1 on chromosome 2p12 is a maternally suppressed gene that is associated paternally with handedness and schizophrenia. *Mol Psychiatry*. 12: 1057,1129–1139.

- 1
2
3 40. Ludwig KU, Mattheisen M, Muhleisen TW, Roeske D, Schmal C, Breuer R, *et al.* (2009):
4 Supporting evidence for LRR1M1 imprinting effects in schizophrenia. *Mol Psychiatry*,
5 2009/07/25. 14: 743–745.
6
7
8
9

10 **Supplementary Figures**

11
12 **Supplementary Figure S1. Distribution of laterality indexes.** Histograms and Q-Q plots for a) PegQ; b)
13 MarkQ; c) SortQ and d) GripQ. Each individual measure is represented by the green rugs at the bottom
14 of each distribution. The blue lines describe normal distributions under the observed mean and
15 variance. The Q-Q plots illustrate the differences between a normal (red lines) and the observed (black
16 circles) distribution.
17

18 **Supplementary Figure S2. Age effect on laterality indexes.** The histograms on the left show the age,
19 measured in days, when data were collected. The graph on the right plot the laterality indexes by
20 age.
21

22 **Supplementary Figure S3. Sex effect of correlation across indexes.** Correlation of laterality measures
23 in A) males and B) females. The cells along the diagonal show the empirical distribution for each index
24 and the bar-plot for the preferred hand for writing. The boxes on the left of the main diagonal show
25 the bivariate distribution of the indexes colour coded for preferred hand for writing (left = orange;
26 right =blue). For example, the first box on the second row of the matrix illustrates the PegQ scores on
27 the x-axis and MarkQ on the y-axis. The bottom row shows the box-plots for each laterality index
28 grouped by preferred hand for writing. The red lines are aligned along the zero for each index. The
29 cells on the right of the diagonal show the correlation coefficients and the sample sizes from which
30 they were calculated. The last column reports the confidence intervals for the box-plots shown in the
31 bottom row, providing a measure on how the different indexes separate the right and left handers for
32 preferred writing hand.
33

34 **Supplementary Figure S4. Performance of both hands.** The performance of both the right and left
35 hand is plotted for each individuals in the four tasks: Pegboard, Marking Squares, Sorting Matches and
36 Grip strength. Equal performance of both hands would plot along the red line.
37
38

39 **Supplementary Figure S5. Age effect on best performance.** The best performance regardless of left
40 or right preference was plotted against age for A) Pegboard, B) Marking squares, C) Sorting matches
41 and D) Grip strength. Correlation and statistical significance are shown in red and a regression line is
42 shown in blue.
43
44
45
46
47
48
49
50
51
52
53
54
55
56
57
58
59
60

Appendix B

Review of “Laterality indexes are poorly correlated but consistently show the tendency of males and females to be more left and right-lateralised, respectively” by Castillo et al. Manuscript submitted for publication in *Royal Society Open Science*.

This manuscript describes a large n study of a few measures of hand performance in groups of right and left handed individuals grouped by hand preference. The authors describe relatively weak correlations between the tests and some small sex differences.

The whole preference versus performance question is quite a big one in handedness research for some time. My own bias is that some of the principal questions asked in that field weren't particularly useful for understanding handedness or brain asymmetry. There! I said it! So now authors and editor you know my own bias, yes?

My recollection of the older stuff is, yes when you measure preference with a questionnaire, you get J shaped distributions and performance you get normal Gaussian distributions shifted towards more right hand skill. Most right handers, but not all, are somewhat better with their right hand. The interesting thing that most people don't comment on is that they are not as much better as you might guess for some tasks, but for one that resemble writing, well yes, of course. Tasks matter and haven't been the subject of much discussion. The second observation is that left handers are not as strongly left handed (not surprising), more of them are better with the right hand than right handers are better with the left hand (not surprising but mechanism unknown). The latter is sometimes explained as a consequence of living in a right handed world, but that is hand waving BALONEY, in my opinion.

From me as a neuropsychologist, all of this is academic unless these distinct measures are yoked to something else of interest. Hand preference, even just for writing, does a not bad job at suggesting unusual cerebral asymmetry but this could be refined. For example, in these kinds of data you might find the thing that is the most right handed in left handers might be a possible predictor of language laterality, as 70% ish of lefties are left hemispheric for language (Carey and Johnsone 2014 Frontiers talks about this).

- 1. The justification of the study needs beefing up, in more than one place. A more complete survey of older claims about performance versus preference would not go amiss. See Bishop (1989) and who ever cites her. Marian Annett's 2000 book has tons of such data and also will summarize some of the older literature.**
- 2. In this older literature somebody, surely looked at sex differences in performance, probably separately in left and right handers.** Review them, critically please and don't just give us some say yes, some say no, further research as needed without a power analysis or some sort of rationale about why this further research helps, even if in a small way.
- 3. Grip strength was done years ago. Only interesting thing about it to me is it shows the unlike what the odd participant tells you, differences in hand strength are small and not very predictive of handedness preference. My**

guess is that distributions won't look very different in right versus left handers. If they do that's interesting.

- 4. In a few places analyses (like the PCA) might be more interesting if done on left handed and right handed writers separately. Too much of this paper is focused on correlations between the four measures, which will always be partly driven by hand preference. If you can't make a better case for this focus, I would think about looking for asymmetrical asymmetries. Their absence or presence is potentially important. My guess is least for grip strength, most for the task that uses the pencil.**

Sorry to be tough, my Scottish colleagues, but I think this could be improved. I look forward to reviewing a slightly beefed up version—convince me that I am wrong about performance versus preference! Let's also see what any other reviewers and the editor have to say. As is my usual practice, I prefer my reviews to remain signed.

David Carey
Bangor University
04/11/2019

Minor comments see annotated pdf for many—often first thing to pop into my head so do with what you will!

Appendix C

Dear Prof Essi Viding,

As requested, we confirm we list in the manuscript the following sections:

- *Ethics statement.* Information about ethical approvals were reported as part of the cohort description. We now report this information in a separate section.
- *Data accessibility.* We have renamed our previous section called “data availability” as “data accessibility”. As we report a secondary analysis in the ALSPAC sample, we are not able to share the data because of the data provider policy. The data can be accessed through a request to the ALSPAC executive. However, we shared the full description of the data used as well as our code, so that our analysis could be reproduced. We report now in the data accessibility section a link to the code.
- *Competing interests.* This has been added.
- *Authors’ contributions.* This has been added.
- *Acknowledgements.* The relevant information is now reported here.
- *Funding statement.* This has been added and it lists the relevant information previously listed under the “Acknowledgements” section.

We confirm that all authors have agreed on this revision.

Two reviewers have provided commentary on your paper, and the general view appears to be the manuscript is heading in the right direction, but requires some work to get it over the line. Please ensure you respond fully to the reviewer comments in both a tracked-change version of the manuscript, and a point-by-point response detailing how you've addressed the reviewers' concerns.

Additionally, we note that, although only one author is listed on the electronic submission form, a number of co-authors are included in the manuscript. Please ensure you list all authors in both the form and the manuscript, and bear in mind that all authors conform to the requirements at <https://royalsociety.org/journals/ethics-policies/openness/>. If you need to make any changes to the authorship, this must be communicated to the editorial office, and ALL authors must agree the changes before they are made.

- RESPONSE TO THE EDITOR:

Thank you for the positive and constructive feedback. We include in our reply a tracked-change version of the revised manuscript and a point-by-point response to the reviewers. The original text from the reviewers is in italics and we started our response with a bullet point. We did our best to address all the minor comments raised in the annotated PDF by Reviewer 1 in a logical way. All authors have added to the online form.

Reviewer: 1

This manuscript describes a large n study of a few measures of hand performance in groups of right and left handed individuals grouped by hand preference. The authors describe relatively weak correlations between the tests and some small sex differences. The whole preference versus performance question is quite a big one in handedness research for some time. My own bias is that some of the principal questions asked in that field weren't particularly useful for understanding handedness or brain asymmetry. There! I said it! So now authors and editor you know my own bias, yes?

My recollection of the older stuff is, yes when you measure preference with a questionnaire, you get J shaped distributions and performance you get normal Gaussian distributions shifted towards more right hand skill.

Most right handers, but not all, are somewhat better with their right hand. The interesting thing that most people don't comment on is that they are not as much better as you might guess for some tasks, but for one that resemble writing, well yes, of course. Tasks matter and haven't been the subject of much discussion.

The second observation is that left handers are not as strongly left handed (not surprising), more of them are better with the right hand than right handers are better with the left hand (not surprising but mechanism unknown). The latter is sometimes explained as a consequence of living in a right handed world, but that is hand waving BALONEY, in my opinion.

From me as a neuropsychologist, all of this is academic unless these distinct measures are yoked to something else of interest. Hand preference, even just for writing, does a not bad job at suggesting unusual cerebral asymmetry but this could be refined. For example, in these kinds of data you might find the thing that is the most right handed in left handers might be a possible predictor of language laterality, as 70% ish of lefties are left hemispheric for language (Carey and Johnstone 2014 Frontiers talks about this).

- We thank the reviewer for such an honest, informed and spontaneous (almost like the Joyce's Ulysses.... There! I said it! And, it is a compliment, of course). The reviewer summarised in two words at the beginning of his review the exact point that we are making with this study "Tasks matter" and indeed this issue haven't been the subject of much discussion. We agree with the reviewer that what we present needs to be useful for something. Here, we would like to emphasise that we are not trying to say that one measure is better than the other, but rather that different measures cannot be used interchangeably. Language lateralization is of obvious interest in the context of handedness research, but equally handedness is studied in the context of genetics, psychiatric disorders, and human evolution. A key issue in this vast and diverse literature is the inconsistency on how handedness is measured and defined. Specifically, while most literature refer to the distinction of preference versus performance, we are arguing that we cannot refer to performance as a general term as there is huge variability across tests. Furthermore, we cannot talk about handedness generally, without specifying the task used for measurements. The key aim of our study is to provide evidence that tasks and

handedness definition are of key importance for reproducibility of findings. By revising the introduction and discussion we hope that the goal of our study is now better defined. In addition, following the reviewer suggestions, we now provide additional analyses for these datasets, looking specifically at the differences between left and right handers. We added in the manuscript analysis of correlations (Supplementary Figure S4), index distributions (Table 3) and hand preference effect on performance (Table 5). Related to this, we also report in the introduction the well-made point by the reviewer that left handers are less consistent than right handers in their performance (pag 3 line 9).

1. The justification of the study needs beefing up, in more than one place. A more complete survey of older claims about performance versus preference would not go amiss. See Bishop (1989) and who ever cites her. Marian Annett's 2000 book has tons of such data and also will summarize some of the older literature.

- The reviewer is absolutely correct in saying that the introduction was not sufficiently developed. We have now extensively revised the introduction and added a completely new section which covers in more details the previous literature to better define the gaps that we are addressing (end of pag 2 – pag 3). As the reviewer points out in the general comments, the discussion around different laterality indexes has not been extensively covered in the literature. Furthermore, sex effects on laterality indexes have not been systematically investigated. A key challenge is the difficulty in collecting multiple measures in the same participants, which we overcame by taking advantage of a large population cohort. The new section in the introduction covers in more details older studies that looked at the relationships across indexes and it spells out the gaps in the literature (i.e. relationship across laterality indexes and sex effects on laterality indexes).

2. In this older literature somebody, surely looked at sex differences in performance, probably separately in left and right handers. Review them, critically please and don't just give us some say yes, some say no, further research as needed without a power analysis or some sort of rationale about why this further research helps, even if in a small way.

- The reviewer makes a valid point in suggesting that sex effects have been looked on performance tasks (performance = speed, strength etc...), which is indeed the case. However, sex effects have not been analysed on laterality indexes derived from performance measures (e.g. PegQ, MarkQ, etc). We agree this was not fully explained in the introduction. We have now expanded this section to highlight this gap in the literature (first paragraph pag 4). We also added references that looked laterality indexes in left and right handers separately to support the new set of analyses suggested by the reviewer. In this case, differences between left and right handers have been reported for laterality indexes, but in very few studies (pag 3 line 23).

3. Grip strength was done years ago. Only interesting thing about it to me is it shows the unlike what the odd participant tells you, differences in hand strength are small and not very predictive of handedness preference. My guess is that distributions won't look very different in right versus left handers. If they do that's interesting.

- Although it is true that grip strength was one of the first tasks used to derive laterality indexes, it is still used in the laterality literature. E.g.: Gumus M. Analysis of the Relationship between Cerebral Lateralization and Grip Strength in Elite Fencing Athletes. *Life Sci J* 2017;14(8):97-104; Yu et al (2017) Degree, but not direction of grip strength asymmetries, is related to depression and anxiety in an elderly population, *Laterality*, 22:3, 268-278; Loprinzi et al Handedness, Grip Strength, and Memory Function: Considerations by Biological Sex *Medicina* 2019, 55(8), 444.

When looking at our results, we found that, even if GripQ is worse than other tasks at separating left and right handers (Figure 1) it still detects a difference between the means of the two populations. We now provide the descriptive statistics for the distributions of the right and left handers (Table 3). Therefore, given the historical perspective and its continuous use, we believe that it is important to include GripQ in this analysis. Furthermore, given that our goal was to analyse as many laterality measures as possible, we included all those available in ALSPAC. This dataset will therefore provide a useful reference to researchers employing different tasks, including grip strength.

4. In a few places analyses (like the PCA) might be more interesting if done on left handed and right handed writers separately. Too much of this paper is focused on correlations between the four measures, which will always be partly driven by hand preference. If you can't make a better case for this focus, I would think about looking for asymmetrical asymmetries. Their absence or presence is potentially important. My guess is least for grip strength, most for the task that uses the pencil.

- The reviewer is correct in saying that the focus of the paper is in looking at the correlation of the indexes (together with sex effects) and we hoped we have now made a stronger case for this analysis in the introduction. The reviewer is also correct in pointing that hand preference is driving the correlations. Our analysis shows that, but it also offers more insights in how different measures relate to each other. We agree with the reviewer that additional analyses would be of interest and, following the suggestions, we now analyse left and right handers separately. We present this analysis for the correlation matrices with the plots shown in supplementary materials and statistics as a new table (Table 3). This analysis shows that although the left and right handers have negative and positive means respectively for all indexes and that the left handers are less consistent, there is variability across indexes. We also analysed left and right handers separately for the PCA. We include the results in our response, but as they provide little information, we preferred to leave them out from the manuscript. The results show that after removing the left/right dimension there are not substantial differences to be

highlighted in the structure of the data compared to what reported in the PCA analysis of the entire samples. This was particularly true for the right handers as dominant group of the dataset. We are happy to include this in the manuscript if it will be requested. Finally, we included hand preference in our analysis of sex effect on performance (Table 5). We carried out an ANOVA showing that sex has a strong effect on performance for all indexes which however is not influenced by hand preference. Following these analyses we added a paragraph to the discussion (pag 18 from line 9).

see review and annotated pdf for minor comments.

- We list here how we addressed all points listed in the annotated PDF.

Abstract

The abstract has been significantly revised. As part of the revision we specifically address the following comments:

Comment 1: We remove the second sentence so that the link between first and third sentence is more obvious.

Comment 2: We used measures previously reported in the literature (including grip) that were available in ALSPAC. We now specify this in the abstract.

Comment 3: The reviewer asked to report more statistics in the paper. We have now changed the narrative of our results and provided more details in the abstract. We prefer avoiding to refer to statistical significance (which indeed is reached in some of our analysis) to be in line with the recommendation in the field <https://www.nature.com/articles/d41586-019-00874-8> . Avoiding mention of statistical significance is also a recommendation by ALSPAC policies.

Comment 4/5: We changed the sentence with “Males were also over-represented among the individuals performing equally with both hands suggesting they had a higher tendency to be weakly lateralised.”

Introduction

The introduction was extensively revised in response to the main points. During this process we took into account all annotated comments. For the first three paragraphs these were mainly about writing style.

Fourth paragraph: here we developed the point about lack of data in the literature looking at the relationship across laterality indexes, rather than the correlation between preference and performance. We completely revised this section to provide more details on the previous literature.

As explained in the second main point, we have now explained that we are looking at sex effect on the laterality indexes (rather than on performance), spelling out that this type of analysis has not been covered in previous research. We still cite at this point Marietta P-P, which we did also in the original submission but was probably not obvious because of the numbered style.

Materials and Methods

For this session the reviewer makes three comments: 1) asking for more details about the tasks; 2) making the data available and 3) a comment about the recording of the dominant/non dominant hand for the different task.

About comment 1), we would like to clarify that we did not carry out the experiments ourselves, so we need to rely on the ALSPAC documentation. We have reported in the methods what we consider to be the key information and we added in the supplementary material the relevant sections taken from the ALSPAC documentation, which is publicly available. For clarity, we have now specified in the text that the pegboard and grip test were collected to assess primarily dexterity and strength, respectively, while the marking square and sorting matches tests were collected specifically to measure laterality.

About comment 2), we cannot share the ALSPAC data directly, but the data can be accessed through a request to ALSPAC Executive. Before submitting the manuscript, we got confirmation from the Journal that this was acceptable. In fact, Royal Society Open Science has published papers using ALSPAC data under the same arrangement.

About comment 3), we are not entirely sure about the point made by the reviewer as we are not referring to mismatches. The issue here is that the derivation of the indexes requires scores for the LEFT and RIGHT hand. These are available for the grip task but not for the other tasks for which the scores were coded as PREFERRED or NON PREFERRED hand for writing. Therefore, we derived the scores for the left and right hand based on the self-reported preferred hand for writing. We hope that the inclusion of additional documentation clarifies this point. The observation that the distributions obtained for the individual tasks are in line with what expected, give us confidence in the process.

Results

First section: the reviewer's comments for the first section appears to be minor, either stylistic or asking for clarifications. In one comment, the reviewer asks if Table 1 is necessary. We preferred to leave Table 1 as we believe it is useful to see the numbers of the study. The reviewer asked for more details about previous work using the marking square test. We have now cover this in the introduction (pag 3 line 14). The point that we are making here in the results is that all the tasks gave the expected distributions based on previous work. MarQ is the only index that shows a bimodal distribution as expected (see reference 16). The reviewer is correct in suggesting that the individuals with negatives MarkQ scores are more likely to be left-handed; this is shown in Figure 1. The final comment for this section is about the (lack of) age effect. This is relevant for the sex analysis: by

presenting lack of age effect we are ruling out that the sex effect might have been influenced by age.

Figure 1/2: The reviewer asks to show that MarkQ separates left and right handers. This is shown in the bottom row of Figure 1. We now reference directly this element of the figure in the text. The reviewer asks to provide correlations across indexes and PCA analysis for left and right handers separately. We have now included both analyses. The results for the correlation analysis in left and right handers are shown in Supplementary Figure S4 and as a new table (Table 3). We run the PCA analysis in left and right handers separately, but we do not think it adds much to the main findings. This is submitted as part of our revision and we are happy to include it in the manuscript, if necessary. Finally, we included hand preference as part of the sex effects analysis on performance (new Table 5). We also fixed the labels in the figure showing the PCA results.

Sex effect/Table 4 (now Table 5): The reviewer had a number of comments on this section. Here we look at effects on performance and therefore choose the best score regardless of which hand was used. Specifically, we are asking the questions “Who is better? Males or females? Left or right handers?”. Other comments referred to what was Table 4, which has now been replaced by a new Table 5 (ANOVA results), discussed in the previous point. This new analysis includes the effect of left/right hand preference requested by the reviewer. One specific comment referred to the justification for using the Welch t-test. Although no longer relevant for Table 5, we used the same test for Table 6. This test is a standard way to assess whether two population have the same means. As it is a standard procedure, we do not feel it needs to be justified in the manuscript.

Discussion: The reviewer asks whether in Table 5 (now Table 6) we control for the number of left-handers. The answer is no, because the point here it to show that regardless of the task (including GripQ which does not do a very good job at separating left and right handers) we observe the same sex effect. On this point, we would like to note that for the analysis presented in Figure 3, which shows the male/female ratio along the distributions of the indexes, we corrected for the overall male/female ratio to control for the excess of left-handers in males.

Reviewer: 2

Comments to the Author(s)

Summary of Study

In this manuscript the authors evaluated the relationship between writing hand preference and its association with four other manual laterality dexterity indexes (pegboard task, marking squares & sorting matches, grip strength). Outcomes suggested there was a right hand performance bias at the population level – however each index had low correlation with other hand indexes. The authors interpreted the results to mean that different hand measures tap different contexts and are not interchangeable. An overarching theme across indexes was that at the population level, females demonstrated a higher rate of right hand

bias than males (across indexes) and that males demonstrated a stronger tendency to be poorly lateralized compared with females.

General Comments:

The subject matter of the study is important and it is imperative that researchers carefully consider the type of test of handedness best suited for their research question because different tests will indeed tap different skills. I also fully agree with the author's statement recommending that researchers "avoid referring to handedness as a generic measure or a universal concept, and encourage instead to refer to the specific tasks used for handedness assessment". However, it is not surprising that these indexes did not all correlate with one another because they range from fine motor to gross motor to hand strength. It would have been interesting for the authors to consider these different types of skill that the tasks were eliciting and make some interpretations of the results based on what we know about brain organization for fine motor sequencing for both speech and hands.

There was the suggestion in the introduction that hand biases are tied to cognition, disorders and evolution, yet the discussion did not elaborate on any of these points. While the cause-effect relationship is debatable, behavioural markers are valuable and could drive additional research. It was unclear if the longitudinal dataset could have included cognitive scores to help address how the different types of hand performance tasks associate with different cognitive skills.

If there is the suggestion that the biases are linked with sex and disorders –as a reader I would like some framing of what the hand performance measures mean – E.g. how do these findings fit within and evolutionary or developmental frame work – or supposition about the brain organization and cognition.

- After considering the comments by both reviewers we significantly revise the introduction and discussion to be more specific in presenting our research questions. We feel that this general revision should address the main comments by the reviewer. More specifically we introduced changes that are in direct response to these comments.
 - The reviewer asked about possible links between handedness and cognitive skills. We rewrote completely the first paragraph of the introduction where links with cognitive, skills were mentioned, to better set the scene and framework of our analysis. The ALSPAC dataset has a large number of cognitive measures and a follow up analysis of the present work is indeed to look for the possible patterns between handedness and cognitive skills. However, the primary goal of the present study was to explore the relationships across handedness measures, building the foundation for future work, therefore we preferred not to diverge too much in this specific direction. In fact, in addition to cognitive skills, ALSPAC has data on personality traits which we would also like also to analyse under the same model. Equally, this analysis of the laterality indexes will support other lines of research, e.g. testing for genetic associations or relationship with particular disorders. Therefore, we preferred to keep separate any kind of follow-up analyses and instead revised the introduction and discussion to better frame the contribution of current our work. The introduction now includes a new

section (from bottom of pag 2) that reviews in more details the previous literature to better illustrate where this work sits.

- The reviewer also makes the very valid point of elaborating on how the different tasks link with different skills and with brain organization and how the findings fit with an evolutionary or developmental framework. We have added a new section in the discussion (pag 17 lines 17-30) which covers more in details the skills linked to the tasks with mention of hemispheric specialization. We appreciate that we probably have not developed these points as much as the reviewer would have liked, but hopefully the revised introduction now explains better our aims and where this work fits. We also feel that, at this stage we cannot read too much into our results as we would risk running into speculations rather than well supported arguments.

Abstract

There are some inconsistencies in the abstract that will confuse the reader. First it is suggested that males are more right biased but later in the manuscript it is reversed to suggest that females are more right biased.

- Thank you for spotting this! We have corrected it now.

Second, the Grip Task is listed as a test of dexterity – where it is a gross motor test of strength.

- We could not identify the section where we listed Grip as a dexterity task in the abstract and throughout the manuscript. We are happy to correct any sentence that might be misleading, but we would need a more precise indication of where changes are required.

Finally, I am not certain that I would agree that performance tests are costly in comparison to other testing methods, but agree that they are time consuming.

- Contrary to self-reported hand preference through a tick-box questionnaire, performance tasks require data acquisition by specialised personnel. Therefore, although the instrumentation required is not expensive, personnel cost can be significant. Given the requirement for one-to-one assessment, large scale collection is impractical. For clarity we now specify this in the introduction by adding the phrase “as they require one-to-one assessment by trained personnel.”

Introduction Page 3 of 17:

Lines 7/8: The rightward prevalence of handedness is a feature specific to humans: This is vague because it does not reference any literature nor does it concede that apes show right

handedness for tool use – although the prevalence/strength of handedness does not reach proportions found in humans (e.g. references).

- The reviewer is absolutely correct on this point, however we were referring more specifically to the 1:9 proportion. We agree this was not clear, so we rephrased as “The 1:9 left to right ratio for hand preference is a feature specific to humans and is observed across populations. Language acquisition, a function characteristic of humans, is also a strongly lateralised trait with left hemisphere dominance observed in most individuals.” See references 1,2 and 3.

Lines 9/10: This statement also needs a reference.

- We agree that a reference is required; we added Corballis 2003.

Lines 11/14: This statement needs some references – and suggests that the study might also consider the evolutionary perspective and/or links between hand dominances and cognition.

- We agree that a reference is required; we added Corballis et al 2012. As explained in the general comments we also extensively revise the introduction and discussion to better contextualise the study.

Lines 18/19: The supposition based on the reference that handedness is a developmental trait tied to language without a consideration that it has an evolutionary history bound to functional brain organisation other than language.

- We have now rewritten and expanded the first paragraph of the introduction to better contextualise this point. The new paragraph now reads:

“ Worldwide, the vast majority of people (roughly 90%) prefer using the right hand for most tasks in contrast to a minority of about 10% who prefer the left hand (1, 2). The 1:9 left to right ratio for hand preference is a feature specific to humans and is observed across populations. Language acquisition, a function characteristic of humans, is also a strongly lateralised trait with left hemisphere dominance observed in most individuals (3). A link between handedness and language, mediated by hemispheric specialization, was first made with the Wada test (4) This link has since provided the foundations for assuming a role of brain asymmetries in human evolution (5). In this context, handedness which is the most obvious and accessible lateralised trait, has been investigated for association with cognitive skills, personality and psychiatric disorders (6–8). However, inconsistent findings have revealed complex patterns and the cause/effect relationship between handedness, brain asymmetries and disorders remains unexplained and debated (9).”

Lines 28/29: The EDI is also context specific – providing only tool using exemplars. Studies that consider hand actions across a variety of context find that dominances shift.

- We edited the section about EHI also in response to the other reviewer. Most importantly, we revised substantially the introduction to discuss in more details the different handedness tasks.

Lines 56/57: Is this the hypothesis? Maybe reframe as the research question? 'E.g. One would assume that...?'

- This particular sentence has been removed in the revision of the introduction which we hope that now generally explains better the aims of the study.

Introduction Page 4 of 17

Lines 3/4: It is agreed that performance tests can be time consuming, but are they considered expensive as a methodological approach in comparison to other experimental testing?

- See point made in the abstract and addressed in introduction. The cost are associated to the requirement of personnel to acquire the scores.

Lines 14/15: In the abstract it reads as if males are more right lateralised and females more left lateralised – but here it is the other way around.

- We corrected this in the abstract

What is the rationale for adding grip strength to a test of hand dominance for dexterity? Why would we expect correlation?

- Grip strength was one of the first tasks used to derive laterality indexes. As our goal was to assess the relationship across different tasks, we used all relevant measures available in the ALSPAC dataset. This is now better explained in the new section that we added in the introduction which also gives additional references of studies that used the grip test. (See also response to the other Reviewer on the same point, where we list a number of recent studies using grip strength to assess laterality)

Lines 38/39: It would be useful if the authors could offer perspective on why the measures do not correlate – what different skills are we tapping – might they be governed by different and/or overlapping mechanisms? Do the measures correlate with cognition?

- This is a valid point, but we could also turn around the question and ask why would one assume that the measures would correlate? Our analysis fills a gap in the literature (= analysis of multiple laterality measures in the same individuals) providing evidence for poor correlation and adding some clarity into the handedness field which is quite messy when it comes to measurement definitions. As already discussed, we have also revised the introduction to make our goal clear upfront. Rather than speculating on the reasons behind the level of correlations, we have expanded the discussion to cover more in detail the differences between tasks and the type of skills required (pag 17 from line 17). In response to the second question

about correlation with cognitive skills, please see our response to the general comments. We hope that the extensive revision of the introduction and discussion make our goals and rationale clearer.

Materials and Methods page 5 of 17

Lines 27: capitalise 'Marking Square and Sorting Matches' is sometimes in capitals and sometimes not – please be consistent.

- Thank you for spotting this, which has been corrected.

Results

It seems likely that the reason that the MarkQ and PegQ were most highly correlated with Hand Writing because they required the fine motor grip type as handwriting – whereas the SortQ was more gross motor and GripQ was about strength and not dexterity. This could be elaborated on in the discussion.

- This point has been addressed by revising the discussion and including a section on the different skills required by the various tasks. This new section not only cover the differenced across tasks but it also offers an explanation for the lack of a universal handedness performance measure (pag 17).